# A Neural Framework for Generalized Causal Sensitivity Analysis

**Dennis Frauen**[1, 2, 6]     **Fergus Imrie**[3]     **Alicia Curth**[4]     **Valentyn Melnychuk**[1, 2]

**Stefan Feuerriegel**[1, 2]          **Mihaela van der Schaar**[4, 5]

## Abstract

Unobserved confounding is common in many applications, making causal inference from observational data challenging. As a remedy, *causal sensitivity analysis* is an important tool to draw causal conclusions under unobserved confounding with mathematical guarantees. In this paper, we propose NeuralCSA, a neural framework for *generalized* causal sensitivity analysis. Unlike previous work, our framework is compatible with (i) a large class of sensitivity models, including the marginal sensitivity model, $f$-sensitivity models, and Rosenbaum's sensitivity model; (ii) different treatment types (i.e., binary and continuous); and (iii) different causal queries, including (conditional) average treatment effects and simultaneous effects on multiple outcomes. The generality of NeuralCSA is achieved by learning a latent distribution shift corresponding to a treatment intervention using two conditional normalizing flows. We provide theoretical guarantees that NeuralCSA can infer valid bounds on the causal query of interest and also demonstrate this empirically using both simulated and real-world data.

## 1 Introduction

Causal inference from observational data is central to many fields such as medicine (Frauen et al., 2023a; Feuerriegel et al., 2024), economics (Imbens & Angrist, 1994), or marketing (Varian, 2016). However, the presence of unobserved confounding often renders causal inference challenging (Pearl, 2009). As an example, consider an observational study examining the effect of smoking on lung cancer risk, where potential confounders, such as genetic factors influencing smoking behavior and cancer risk (Erzurumluoglu & et al., 2020), are not observed. Then, the causal relationship is not identifiable, and point identification without additional assumptions is impossible (Pearl, 2009).

Causal sensitivity analysis offers a remedy by moving from *point* identification to *partial* identification. To do so, approaches for causal sensitivity analysis first impose assumptions on the strength of unobserved confounding through so-called sensitivity models (Rosenbaum, 1987; Imbens, 2003) and then obtain bounds on the causal query of interest. Such bounds often provide insights that the causal quantities can not reasonably be explained away by unobserved confounding, which is sufficient for consequential decision-making in many applications (Kallus et al., 2019).

Existing works on causal sensitivity analysis can be loosely grouped by problem settings. These vary across (1) sensitivity models, such as the marginal sensitivity model (MSM) (Tan, 2006), $f$-sensitivity model (Jin et al., 2022), and Rosenbaum's sensitivity model (Rosenbaum, 1987); (2) treatment type (i.e., binary and continuous); and (3) causal query of interest. Causal queries may include (conditional) average treatment effects (CATE), but also distributional effects or simultaneous effects on multiple outcomes. Existing works typically focus on a specific sensitivity model, treatment type, and causal query (Table 1). However, none is applicable to all settings within (1)–(3).

---

[1] LMU Munich  [2] Munich Center for Machine Learning  [3] UCLA  [4] University of Cambridge  [5] Alan Turing Institute          [6] Corresponding author (`frauen@lmu.de`)

To fill this gap, we propose NEURALCSA, a neural framework for causal sensitivity analysis that is applicable to numerous sensitivity models, treatment types, and causal queries, including multiple outcome settings. For this, we define a large class of sensitivity models, which we call *generalized treatment sensitivity models* (GTSMs). GTSMs include common sensitivity models such as the MSM, $f$-sensitivity models, and Rosenbaum's sensitivity model. The intuition behind GTSMs is as follows: when intervening on the treatment $A$, the $U$–$A$ edge is removed in the corresponding causal graph, which leads to a distribution shift in the latent confounders $U$ (see Fig. 1). GTSMs then impose restrictions on this latent distribution shift, which corresponds to assumptions on the "strength" of unobserved confounding.

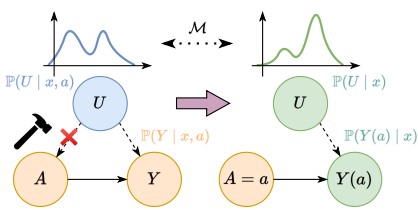

Figure 1: Idea behind NEURALCSA to learn the latent distribution shift due to treatment intervention (↗). Orange nodes denote observed (random) variables. Blue nodes denote unobserved variables pre-intervention. Green nodes indicate unobserved variables post-intervention under a GTSM $\mathcal{M}$. Observed confounders $X$ are empty for simplicity.

NEURALCSA is compatible with any sensitivity model that can be written as a GTSM. This is crucial in practical applications, where sensitivity models correspond to different assumptions on the data-generating process and may lead to different results (Yin et al., 2022). To achieve this, NEURALCSA *learns the latent distribution shift in the unobserved confounders* from Fig. 1 using two separately trained conditional normalizing flows (CNFs). This is different from previous works for causal sensitivity analysis, which do not provide a unified approach across numerous sensitivity models, treatment types, and causal queries. We provide theoretical guarantees that NEURALCSA learns valid bounds on the causal query of interest and demonstrate this empirically.

Our **contributions**[7] are: (1) We define a general class of sensitivity models, called GTSMs. (2) We propose NEURALCSA, a neural framework for causal sensitivity analysis under any GTSMs. NEURALCSA is compatible with various sensitivity models, treatment types, and causal queries. In particular, NEURALCSA is applicable in settings for which bounds are not analytically tractable and no solutions exist yet. (3) We provide theoretical guarantees that NEURALCSA learns valid bounds on the causal query of interest and demonstrate the effectiveness of our framework empirically.

## 2 RELATED WORK

In the following, we provide an overview of related literature on partial identification and causal sensitivity analysis. A more detailed overview, including literature on point identification and estimation, can be found in Appendix A.

**Partial identification:** The aim of partial identification is to compute bounds on causal queries whenever point identification is not possible, such as under unobserved confounding (Manski, 1990). There are several literature streams that impose different assumptions on the data-generating process in order to obtain informative bounds. One stream addresses partial identification for general causal graphs with discrete variables (Duarte et al., 2023).

Another stream assumes the existence of valid instrumental variables (Gunsilius, 2020; Kilbertus et al., 2020). Recently, there has been a growing interest in using neural networks for partial identification (Xia et al., 2021; 2023; Padh et al., 2023). However, none of these methods allow for incorporating sensitivity models and sensitivity analysis.

Table 1: Overview of key settings for causal sensitivity analyses and whether covered by existing literature (✓) or not (✗). Treatments are either binary or continuous. Details are in Appendix A. Our NEURALCSA framework is applicable in all settings.

| Sensitivity model
Causal query | MSM | | $f$-sensitivity | | Rosenbaum | |
|---|---|---|---|---|---|---|
| | Binary | Cont.[†] | Binary | Cont. | Binary | Cont. |
| CATE | ✓ | ✓ | ✓ | ✗ | ✓ | ✗ |
| Distributional effects | ✓ | ✓ | ✗ | ✗ | ✗ | ✗ |
| Interventional density | ✓ | ✓ | (✓) | ✗ | ✗ | ✗ |
| Multiple outcomes | ✗ | ✗ | ✗ | ✗ | ✗ | ✗ |

[†] The MSM for continuous treatment is also called continuous MSM (CMSM) (Jesson et al., 2022).

---

[7] Code is available at https://github.com/DennisFrauen/NeuralCSA.

**Causal sensitivity analysis:** Causal sensitivity analysis addresses the partial identification of causal queries by imposing assumptions on the strength of unobserved confounding via sensitivity models. It dates back to Cornfield et al. (1959), who showed that unobserved confounding could not reasonably explain away the observed effect of smoking on lung cancer risk.

Existing works can be grouped along three dimensions: (1) the sensitivity model, (2) the treatment type, and (3) the causal query of interest (see Table 1; details in Appendix A). Popular sensitivity models include Rosenbaum's sensitivity model (Rosenbaum, 1987), the marginal sensitivity model (MSM) (Tan, 2006), and $f$-sensitivity models (Jin et al., 2022). Here, most methods have been proposed for binary treatments and conditional average treatment effects (Kallus et al., 2019; Zhao et al., 2019; Jesson et al., 2021; Dorn & Guo, 2022; Dorn et al., 2022; Oprescu et al., 2023). Extensions under the MSM have been proposed for continuous treatments (Jesson et al., 2022; Marmarelis et al., 2023a) and individual treatment effects (Yin et al., 2022; Jin et al., 2023; Marmarelis et al., 2023b). However, approaches for many settings are still missing (shown by ✗ in Table 1). In an attempt to generalize causal sensitivity analysis, Frauen et al. (2023b) provided bounds for different treatment types (i.e., binary, continuous) and causal queries (e.g., CATE, distributional effects but not multiple outcomes). Yet, the results are limited to MSM-type sensitivity models.

To the best of our knowledge, no previous work proposes a unified solution for obtaining bounds under various sensitivity models (e.g., MSM, $f$-sensitivity, Rosenbaum's), treatment types (i.e., binary and continuous), and causal queries (e.g., CATE, distributional effects, interventional densities, and simultaneous effects on multiple outcomes).

## 3 MATHEMATICAL BACKGROUND

**Notation:** We denote random variables $X$ as capital letters and their realizations $x$ in lowercase. We further write $\mathbb{P}(x)$ for the probability mass function if $X$ is discrete, and for the probability density function with respect to the Lebesque measure if $X$ is continuous. Conditional probability mass functions/ densities $\mathbb{P}(Y = y \mid X = x)$ are written as $\mathbb{P}(y \mid x)$. Finally, we denote the conditional distribution of $Y \mid X = x$ as $\mathbb{P}(Y \mid x)$ and its expectation as $\mathbb{E}[Y \mid x]$.

### 3.1 PROBLEM SETUP

**Data generating process:** We consider the standard setting for (static) treatment effect estimation under unobserved confounding (Dorn & Guo, 2022). That is, we have observed confounders $X \in \mathcal{X} \subseteq \mathbb{R}^{d_x}$, unobserved confounders $U \in \mathcal{U} \subseteq \mathbb{R}^{d_u}$, treatments $A \in \mathcal{A} \subseteq \mathbb{R}^{d_a}$, and outcomes $Y \in \mathcal{Y} \subseteq \mathbb{R}^{d_y}$. Note that we allow for (multiple) discrete or continuous treatments and multiple outcomes, i.e., $d_a, d_y \geq 1$. The underlying causal graph is shown in Fig. 2. We have access to an observational dataset $\mathcal{D} = (x_i, a_i, y_i)_{i=1}^n$ sampled i.i.d. from the observational distribution $(X, A, Y) \sim \mathbb{P}_{\mathrm{obs}}$. The full distribution $(X, U, A, Y) \sim \mathbb{P}$ is unknown.

We use the potential outcomes framework to formalize the causal inference problem (Rubin, 1974) and denote $Y(a)$ as the potential outcome when intervening on the treatment and setting it to $A = a$. We impose the following standard assumptions (Dorn & Guo, 2022).

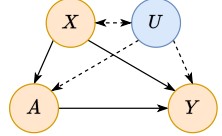

**Assumption 1.** *We assume that for all $x \in \mathcal{X}$ and $a \in \mathcal{A}$ the following three conditions hold: (i) $A = a$ implies $Y(a) = Y$(consistency); (ii) $\mathbb{P}(a \mid x) > 0$ (positivity); and (iii) $Y(a) \perp\!\!\!\perp A \mid X, U$ (latent unconfoundedness).*

Figure 2: Causal graph. Observed variables are colored orange and unobserved blue. We allow for arbitrary dependence between $X$ and $U$.

**Causal queries:** We are interested in a wide range of general causal queries. We formalize them as functionals $Q(x, a, \mathbb{P}) = \mathcal{F}(\mathbb{P}(Y(a) \mid x))$, where $\mathcal{F}$ is a functional that maps the potential outcome distribution $\mathbb{P}(Y(a) \mid x)$ to a real number (Frauen et al., 2023b). Thereby, we cover various queries from the causal inference literature. For example, by setting $\mathcal{F} = \mathbb{E}[\cdot]$, we obtain the conditional expected potential outcomes/ dose-response curves $Q(x, a, \mathbb{P}) = \mathbb{E}[Y(a) \mid x]$. We can also obtain distributional versions of these queries by setting $\mathcal{F}$ to a quantile instead of the expectation. Furthermore, our methodology will also apply to queries that can be obtained by averaging or taking differences. For binary treatments $A \in \{0, 1\}$, the query $\tau(x) = \mathbb{E}[Y(1) \mid x] - \mathbb{E}[Y(0) \mid x]$

is called the conditional average treatment effect (CATE), and its averaged version $\int \tau(x)\mathbb{P}(x)\,\mathrm{d}x$ the average treatment effect (ATE).

Our formalization also covers simultaneous effects on multiple outcomes (i.e., $d_y \geq 2$). Consider query $Q(x, a, \mathbb{P}) = \mathbb{P}(Y(a) \in \mathcal{S} \mid x)$, which is the probability that the outcome $Y(a)$ is contained in some set $\mathcal{S} \subseteq \mathcal{Y}$ after intervening on the treatment. For example, consider two potential outcomes $Y_1(a)$ and $Y_2(a)$ denoting blood pressure and heart rate, respectively. We then might be interested in $\mathbb{P}(Y_1(a) \leq t_1, Y_2(a) \leq t_2 \mid x)$, where $t_1$ and $t_2$ are critical threshold values (see Sec. 6).

## 3.2 CAUSAL SENSITIVITY ANALYSIS

Causal sensitivity analysis builds upon sensitivity models that restrict the possible strength of unobserved confounding (e.g., Rosenbaum & Rubin, 1983a). Formally, we define a sensitivity model as a family of distributions of $(X, U, A, Y)$ that induce the observational distribution $\mathbb{P}_{\mathrm{obs}}$.

**Definition 1.** *A sensitivity model $\mathcal{M}$ is a family of probability distributions $\mathbb{P}$ defined on $\mathcal{X} \times \mathcal{U} \times \mathcal{A} \times \mathcal{Y}$ for arbitrary finite-dimensional $\mathcal{U}$ so that $\int_{\mathcal{U}} \mathbb{P}(x, u, a, y)\,\mathrm{d}u = \mathbb{P}_{\mathrm{obs}}(x, a, y)$ for all $\mathbb{P} \in \mathcal{M}$.*

**Task:** Given a sensitivity model $\mathcal{M}$ and an observational distribution $\mathbb{P}_{\mathrm{obs}}$, the aim of causal sensitivity analysis is to solve the partial identification problem

$$Q_{\mathcal{M}}^+(x, a) = \sup_{\mathbb{P} \in \mathcal{M}} Q(x, a, \mathbb{P}) \quad \text{and} \quad Q_{\mathcal{M}}^-(x, a) = \inf_{\mathbb{P} \in \mathcal{M}} Q(x, a, \mathbb{P}). \tag{1}$$

By its definition, the interval $[Q_{\mathcal{M}}^-(x, a), Q_{\mathcal{M}}^+(x, a)]$ is the tightest interval that is guaranteed to contain the ground-truth causal query $Q(x, a, \mathbb{P})$ while satisfying the sensitivity constraints. We can also obtain bounds for averaged causal queries and differences via $\int Q_{\mathcal{M}}^+(x, a)\mathbb{P}(x)\,\mathrm{d}x$ and $Q_{\mathcal{M}}^+(x, a_1) - Q_{\mathcal{M}}^-(x, a_2)$ (see Appendix D for details).

**Sensitivity models from the literature:** We now recap three types of prominent sensitivity models from the literature, namely, the MSM, $f$-sensitivity models, and Rosenbaum's sensitivity model. These are designed for binary treatments $A \in \{0, 1\}$. To formalize them, we first define the odds ratio $\mathrm{OR}(a, b) = \frac{a}{(1-a)} \frac{(1-b)}{b}$, the observed propensity score $\pi(x) = \mathbb{P}(A = 1 \mid x)$, and the full propensity score $\pi(x, u) = \mathbb{P}(A = 1 \mid x, u)$.[8] Then, the definitions are:

1. The *marginal sensitivity model* (MSM) (Tan, 2006) is defined as the family of all $\mathbb{P}$ that satisfy $\frac{1}{\Gamma} \leq \mathrm{OR}(\pi(x), \pi(x, u)) \leq \Gamma$ for all $x \in \mathcal{X}$ and $u \in \mathcal{U}$ and a sensitivity parameter $\Gamma \geq 1$.

2. $f$-*sensitivity models* (Jin et al., 2022) build upon a given a convex function $f \colon \mathbb{R}_{>0} \to \mathbb{R}$ with $f(1) = 0$ and are defined via $\max \big\{ \int_{\mathcal{U}} f\left(\mathrm{OR}(\pi(x), \pi(x, u))\right) \mathbb{P}(u \mid x, A = 1)\,\mathrm{d}u, \int_{\mathcal{U}} f\left(\mathrm{OR}^{-1}(\pi(x), \pi(x, u))\right) \mathbb{P}(u \mid x, A = 1)\,\mathrm{d}u \big\} \leq \Gamma$ for all $x \in \mathcal{X}$.

3. *Rosenbaum's sensitivity model* (Rosenbaum, 1987) is defined via $\frac{1}{\Gamma} \leq \mathrm{OR}(\pi(x, u_1), \pi(x, u_2)) \leq \Gamma$ for all $x \in \mathcal{X}$ and $u_1, u_2 \in \mathcal{U}$.

**Interpretation and choice of $\Gamma$:** In the above sensitivity models, the sensitivity parameter $\Gamma$ controls the strength of unobserved confounding. Both MSM and Rosenbaum's sensitivity model bound on odds-ratio uniformly over all $u \in \mathcal{U}$, while the $f$-sensitivity model bounds an integral over $u$. We refer to Appendix C for further differences. Setting $\Gamma = 1$ in the above sensitivity models corresponds to unconfoundedness and thus point identification. For $\Gamma > 1$, point identification is not possible, and we need to solve the partial identification problem from Eq. (1) instead.

In practice, one typically chooses $\Gamma$ by domain knowledge or data-driven heuristics (Kallus et al., 2019; Hatt et al., 2022). For example, a common approach in practice is to determine the smallest $\Gamma$ so that the partially identified interval $[Q_{\Gamma}^-(x, a), Q_{\Gamma}^+(x, a)]$ includes 0. Then, $\Gamma$ can be interpreted as a level of "causal uncertainty", quantifying the smallest violation of unconfoundedness that would explain away the causal effect (Jesson et al., 2021; Jin et al., 2023).

---

[8] Corresponding sensitivity models for continuous treatments can be defined by replacing the odds ratio with the density ratio $\mathrm{DR}(a, b) = \frac{a}{b}$ and the propensity scores with the densities $\mathbb{P}(a \mid x)$ and $\mathbb{P}(a \mid x, u)$ (Bonvini et al., 2022; Jesson et al., 2022). We refer to Appendix C for details and further examples of sensitivity models.

## 4 THE GENERALIZED TREATMENT SENSITIVITY MODEL (GTSM)

We now define our generalized treatment sensitivity model (GTSM). The GTSM subsumes a large class of sensitivity models and includes MSM, $f$-sensitivity, and Rosenbaum's sensitivity model).

**Motivation:** Intuitively, we define the GTSM so that it includes all sensitivity models that restrict the latent distribution shift in the confounding space due to the treatment intervention (see Fig. 1). To formalize this, we can write the observational outcome density under Assumption 1 as

$$\mathbb{P}_{\mathrm{obs}}(y \mid x, a) = \int \mathbb{P}(y \mid x, u, a)\, \mathbb{P}(u \mid x, a)\, \mathrm{d}u. \tag{2}$$

When intervening on the treatment, we remove the $U$–$A$ edge in the corresponding causal graph (Fig. 1) and thus artificially remove dependence between $U$ and $A$. Formally, we can write the potential outcome density under Assumption 1 as

$$\mathbb{P}(Y(a) = y \mid x) = \int \mathbb{P}(Y(a) = y \mid x, u)\mathbb{P}(u \mid x)\, \mathrm{d}u = \int \mathbb{P}(y \mid x, u, a)\mathbb{P}(u \mid x)\, \mathrm{d}u. \tag{3}$$

Eq. (2) and (3) imply that $\mathbb{P}_{\mathrm{obs}}(y \mid x, a)$ and $\mathbb{P}(Y(a) = y \mid x)$ only differ by the densities $\mathbb{P}(u \mid x, a)$ and $\mathbb{P}(u \mid x)$ under the integrals (colored red and orange). If the distributions $\mathbb{P}(U \mid x, a)$ and $\mathbb{P}(U \mid x)$ would coincide, it would hold that $\mathbb{P}(Y(a) = y \mid x) = \mathbb{P}_{\mathrm{obs}}(y \mid x, a)$ and the potential outcome distribution would be identified. This suggests that we should define sensitivity models by measuring deviations from unconfoundedness via the shift between $\mathbb{P}(U \mid x, a)$ and $\mathbb{P}(U \mid x)$.

**Definition 2.** *A generalized treatment sensitivity model (GTSM) is a sensitivity model $\mathcal{M}$ that contains all probability distributions $\mathbb{P}$ that satisfy $\mathcal{D}_{x,a}\left(\mathbb{P}(U \mid x), \mathbb{P}(U \mid x, a)\right) \leq \Gamma$ for a functional of distributions $\mathcal{D}_{x,a}$, a sensitivity parameter $\Gamma \in \mathbb{R}_{\geq 0}$, and all $x \in \mathcal{X}$ and $a \in \mathcal{A}$.*

**Lemma 1.** *The MSM, the $f$-sensitivity model, and Rosenbaum's sensitivity model are GTSMs.*

The class of all GTSMs is still too large for meaningful sensitivity analysis. This is because the sensitivity constraint may not be invariant w.r.t. transformations (e.g., scaling) of the latent space $\mathcal{U}$.

**Definition 3** (Transformation-invariance)**.** *A GTSM $\mathcal{M}$ is transformation-invariant if it satisfies $\mathcal{D}_{x,a}(\mathbb{P}(U \mid x), \mathbb{P}(U \mid x, a)) \geq \mathcal{D}_{x,a}(\mathbb{P}(t(U) \mid x), \mathbb{P}(t(U) \mid x, a))$ for any measurable function $t \colon \mathcal{U} \to \widetilde{\mathcal{U}}$ to another latent space $\widetilde{\mathcal{U}}$.*

Transformation-invariance is necessary for meaningful sensitivity analysis because it implies that once we choose a latent space $\mathcal{U}$ and a sensitivity parameter $\Gamma$, we cannot find a transformation to another latent space $\widetilde{\mathcal{U}}$ so that the induced distribution on $\widetilde{\mathcal{U}}$ violates the sensitivity constraint. All sensitivity models we consider in this paper are transformation-invariant, as stated below.

**Lemma 2.** *The MSM, $f$-sensitivity models, and Rosenbaum's sensitivity model are transformation-invariant.*

## 5 NEURAL CAUSAL SENSITIVITY ANALYSIS

We now introduce our neural approach to causal sensitivity analysis as follows. First, we simplify the partial identification problem from Eq. (1) under a GTSM and propose a (model-agnostic) two-stage procedure (Sec. 5.1). Then, we provide theoretical guarantees for our two-stage procedure (Sec. 5.2). Finally, we instantiate our neural framework called NEURALCSA (Sec. 5.3).

### 5.1 SENSITIVITY ANALYSIS UNDER A GTSM

**Motivation:** Recall that, by definition, a GTSM imposes constraints on the distribution shift in the latent confounders due to treatment intervention (Fig. 1). Our idea is to propose a two-stage procedure, where Stage 1 learns the observational distribution (Fig. 1, left), while Stage 2 learns the shifted distribution of $U$ after intervening on the treatment under a GTSM (Fig. 1, right). In Sec. 5.2, we will see that, under weak assumptions, learning this distribution shift in separate stages is guaranteed to lead to the bounds $Q_{\mathcal{M}}^+(x, a)$ and $Q_{\mathcal{M}}^-(x, a)$. To formalize this, we start by simplifying the partial identification problem from Eq. (1) for a GTSM $\mathcal{M}$.

**Simplifying Eq.** (1)**:** We begin by rewriting Eq. (1) using the GTSM definition. Without loss of generality, we consider the upper bound $Q_{\mathcal{M}}^+(x, a)$. Recall that Eq. (1) seeks to maximize over all probability distributions that are compatible both with the observational data and with the sensitivity model. However, note that any GTSM only restricts the $U$–$A$ part of the distribution, not the $U$–$Y$ part. Hence, we can use Eq. (3) and Eq. (2) to write the upper bound as

$$Q_{\mathcal{M}}^+(x, a) = \sup_{\substack{\{\mathbb{P}(U|x,a')\}_{a' \neq a} \\ \text{s.t. } \mathcal{D}_{x,a}(\mathbb{P}(U|x), \mathbb{P}(U|x,a)) \leq \Gamma \\ \text{and } \mathbb{P}(u|x) = \int \mathbb{P}(u|x,a)\mathbb{P}_{\text{obs}}(a|x)\,\mathrm{d}a}} \sup_{\substack{\mathbb{P}(U|x,a), \{\mathbb{P}(Y|x,u,a)\}_{u \in \mathcal{U}} \\ \text{s.t. Eq. (2) holds}}} \mathcal{F}\left(\int \mathbb{P}(Y \mid x, u, a)\mathbb{P}(u \mid x)\,\mathrm{d}u\right), \quad (4)$$

where we maximize over (families of) probability distributions $\{\mathbb{P}(U \mid x, a')\}_{a' \neq a}$ (left supremum), and $\mathbb{P}(U \mid x, a)$, $\{\mathbb{P}(Y \mid x, u, a)\}_{u \in \mathcal{U}}$ (right supremum). The coloring indicates the components that appear in the causal query/objective. The constraint in the right supremum ensures that the respective components of the full distribution $\mathbb{P}$ are compatible with the observational data, while the constraints in the left supremum ensure that the respective components are compatible with both observational data and the sensitivity model.

The partial identification problem from Eq. (4) is still hard to solve as it involves two nested constrained optimization problems. However, we can further simplify Eq. (4): We will show in Sec. 5.2 that we can replace the right supremum with fixed distributions $\mathbb{P}^*(U \mid x, a)$ and $\mathbb{P}^*(Y \mid x, a, u)$ for all $u \in \mathcal{U} \subseteq \mathbb{R}^{d_y}$ so that Eq. (2) holds. Then, Eq. (4) reduces to a single constrained optimization problem (left supremum). Moreover, we will show that we can choose $\mathbb{P}^*(Y \mid x, a, u) = \delta(Y - f_{x,a}^*(u))$ as a delta-distribution induced by an invertible function $f_{x,a}^*: \mathcal{U} \to \mathcal{Y}$. The con-

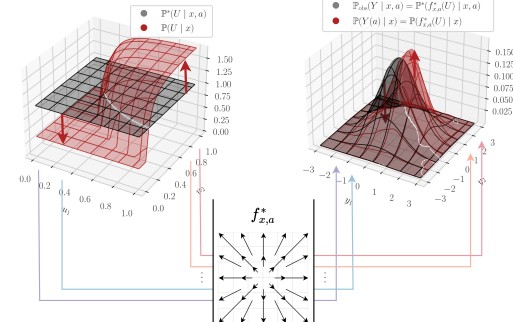

Figure 3: Overview of the two-stage procedure.

straint in Eq. (2) that ensures compatibility with the observational data then reduces to $\mathbb{P}_{\text{obs}}(Y \mid x, a) = \mathbb{P}^*(f_{x,a}^*(U) \mid x, a)$. This motivates the following two-stage procedure (see Fig. 3).

**Two-stage procedure:** In **Stage 1**, we fix $\mathbb{P}^*(U \mid x, a)$ and fix an invertible function $f_{x,a}^*: \mathcal{U} \to \mathcal{Y}$ so that $\mathbb{P}_{\text{obs}}(Y \mid x, a) = \mathbb{P}^*(f_{x,a}^*(U) \mid x, a)$ holds. That is, the induced push-forward distribution of $\mathbb{P}^*(U \mid x, a)$ under $f_{x,a}^*$ must coincide with the observational distribution $\mathbb{P}_{\text{obs}}(Y \mid x, a)$. The existence of such a function is always guaranteed (Chen & Gopinath, 2000). In **Stage 2**, we then set $\mathbb{P}(U \mid x, a) = \mathbb{P}^*(U \mid x, a)$ and $\mathbb{P}(Y \mid x, a, u) = \mathbb{P}^*(Y \mid x, a, u)$ in Eq. (4) and only optimize over the left supremum. That is, we write stage 2 for discrete treatments as

$$\sup_{\substack{\mathbb{P}(u|x, A \neq a) \\ \text{s.t. } \mathbb{P}(u|x) = \mathbb{P}^*(u|x,a)\mathbb{P}_{\text{obs}}(a|x) + \mathbb{P}(u|x, A \neq a)(1 - \mathbb{P}_{\text{obs}}(a|x)) \\ \text{and } \mathcal{D}_{x,a}(\mathbb{P}(U|x), \mathbb{P}^*(U|x,a)) \leq \Gamma}} \mathcal{F}\left(\mathbb{P}(f_{x,a}^*(U) \mid x)\right), \quad (5)$$

where we maximize over the distribution $\mathbb{P}(u \mid x, A \neq a)$ for a fixed treatment intervention $a$. For continuous treatments, we can directly take the supremum over $\mathbb{P}(u \mid x)$.

## 5.2 Theoretical guarantees

We now provide a formal result that our two-stage procedure returns valid solutions to the partial identification problem from Eq. (4). The following theorem states that Stage 2 of our procedure is able to attain the optimal upper bound $Q_{\mathcal{M}}^+(x, a)$ from Eq. (4), even after fixing the distributions $\mathbb{P}^*(U \mid x, a)$ and $\mathbb{P}^*(Y \mid x, a, u)$ as done in Stage 1. A proof is provided in Appendix B.

**Theorem 1** (Sufficiency of two-stage procedure). *Let $\mathcal{M}$ be a transformation-invariant GTSM. For fixed $x \in \mathcal{X}$ and $a \in \mathcal{A}$, let $\mathbb{P}^*(U \mid x, a)$ be a fixed distribution on $\mathcal{U} = \mathbb{R}^{d_y}$ and $f_{x,a}^*: \mathcal{U} \to \mathcal{Y}$ a fixed invertible function so that $\mathbb{P}_{\text{obs}}(Y \mid x, a) = \mathbb{P}^*(f_{x,a}^*(U) \mid x, a)$. Let $\mathcal{P}^*$ denote the space of all full probability distributions $\mathbb{P}^*$ that induce $\mathbb{P}^*(U \mid x, a)$ and $\mathbb{P}^*(Y \mid x, u, a) = \delta(Y - f_{x,a}^*(u))$ and that satisfy $\mathbb{P}^* \in \mathcal{M}$. Then, under Assumption 1, it holds that $Q_{\mathcal{M}}^+(x, a) = \sup_{\mathbb{P}^* \in \mathcal{P}^*} Q(x, a, \mathbb{P}^*)$ and $Q_{\mathcal{M}}^-(x, a) = \inf_{\mathbb{P}^* \in \mathcal{P}^*} Q(x, a, \mathbb{P}^*)$.*

**Intuition:** Theorem 1 has two major implications: (i) It is sufficient to fix the distributions $\mathbb{P}^*(U \mid x, a)$ and $\mathbb{P}^*(Y \mid x, u, a)$, i.e., the components in the right supremum of Eq. (4) and only optimize over the left supremum; and (ii) it is sufficient to choose $\mathbb{P}^*(Y \mid x, u, a) = \delta(Y - f^*_{x,a}(u))$ as a delta-distribution induced by an invertible function $f^*_{x,a} : \mathcal{U} \to \mathcal{Y}$, which satisfies the data-compatibility constraint $\mathbb{P}_{\text{obs}}(Y \mid x, a) = \mathbb{P}^*(f^*_{x,a}(U) \mid x, a)$. **Intuition for (i):** In Eq. (4), we optimize jointly over all components of the full distribution. This suggests that there are multiple solutions that differ only in the components of unobserved parts of $\mathbb{P}$ (i.e., in $\mathcal{U}$) but lead to the same potential outcome distribution and causal query. Theorem 1 states that we may restrict the space of possible solutions by fixing the components $\mathbb{P}^*(U \mid x, a)$ and $\mathbb{P}^*(Y \mid x, a, u)$, without loosing the ability to attain the optimal upper bound $Q^+_{\mathcal{M}}(x, a)$ from Eq. (4). **Intuition for (ii):** We cannot pick *any* $\mathbb{P}^*(Y \mid x, a, u)$ that satisfies Eq. (2). For example, any distribution that induces $Y \perp\!\!\!\perp U \mid X, A$ would satisfy Eq. (2), but implies unconfoundedness and would thus not lead to a valid upper bound $Q^+_{\mathcal{M}}(x, a)$. Intuitively, we have to choose a $\mathbb{P}(Y \mid x, a, u)$ that induces "maximal dependence" (mutual information) between $U$ and $Y$ (conditioned on $X$ and $A$), because the GTSM does not restrict this part of the full probability distribution $\mathbb{P}$. The maximal mutual information is achieved if we choose $\mathbb{P}(Y \mid x, a, u) = \delta(Y - f^*_{x,a}(u))$.

### 5.3 NEURAL INSTANTIATION: NEURALCSA

We now provide a neural instantiation called NEURALCSA for the above two-stage procedure using conditional normalizing flows (CNFs) (Winkler et al., 2019). The architecture of NEU-RALCSA is shown in Fig. 4. NEURALCSA instantiates the two-step procedure as follows:

**Stage 1:** We fix $\mathbb{P}^*(U \mid x, a)$ to the standard normal distribution on $\mathcal{U} = \mathbb{R}^{d_y}$. Our task is then to learn an invertible function $f^*_{x,a} : \mathcal{U} \to \mathcal{Y}$ that maps the standard Gaussian distribution on $\mathcal{U}$ to $\mathbb{P}_{\text{obs}}(Y \mid x, a)$. We model $f^*_{x,a}$ as a CNF $f^*_{g^*_\theta(x,a)}$, where $f^*$ is a normalizing flow (Rezende & Mohamed, 2015), for which the parameters are the output of a fully connected

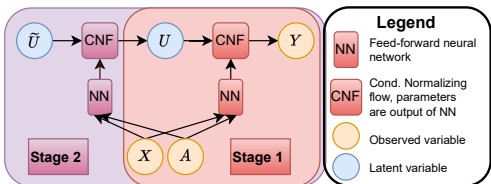

Figure 4: Architecture of NEURALCSA.

neural network $g^*_\theta$, which itself is parametrized by $\theta$ (Winkler et al., 2019). We obtain $\theta$ by maximizing the empirical Stage 1 loss $\mathcal{L}_1(\theta) = \sum_{i=1}^n \log \mathbb{P}(f^*_{g^*_\theta(x_i, a_i)}(U) = y_i)$, where $U \sim \mathcal{N}(0_{d_y}, I_{d_y})$ is standard normally distributed. The stage 1 loss can be computed analytically via the change-of-variable formula (see Appendix F).

**Stage 2:** In Stage 2, we need to maximize over distributions on $U$ in the latent space $\mathcal{U}$ that maximize the causal query $\mathcal{F}(\mathbb{P}(f^*_{g^*_{\theta_{\text{opt}}}(x,a)}(U) \mid x))$, where $\theta_{\text{opt}}$ is a solution from maximizing $\mathcal{L}_1(\theta)$ in stage 1. We can do this by learning a second CNF $\widetilde{f}_{\widetilde{g}_\eta(x,a)}$, where $\widetilde{f} : \widetilde{\mathcal{U}} \to \mathcal{U}$ is a normalizing flow that maps a standard normally distributed auxiliary $\widetilde{U} \sim \mathcal{N}(0_{d_y}, I_{d_y})$ to the latent space $\mathcal{U}$, and whose parameters are the output of a fully connected neural network $\widetilde{g}_\eta$ parametrized by $\eta$. The CNF $\widetilde{f}_{\widetilde{g}_\eta(x,a)}$ from Stage 2 induces a new distribution on $U$, which mimics the shift due to unobserved confounding when intervening instead of conditioning (i.e., going from Eq. (2) to Eq. (3)). We can compute the query under the shifted distribution by concatenating the Stage 2 CNF with the Stage 1 CNF and applying $\mathcal{F}$ to the shifted outcome distribution (see Fig. 4). More precisely, we optimize $\eta$ by maximizing or minimizing the empirical Stage 2 loss

$$\mathcal{L}_2(\eta) = \sum_{i=1}^n \mathcal{F}\left(\mathbb{P}\left(f^*_{g^*_{\theta_{\text{opt}}}(x_i, a_i)}\left((1 - \xi_{x_i, a_i})\widetilde{f}_{\widetilde{g}_\eta(x_i, a_i)}(\widetilde{U}) + \xi_{x_i, a_i}\widetilde{U}\right)\right)\right), \quad (6)$$

where $\xi_{x_i, a_i} = \mathbb{P}_{\text{obs}}(a_i \mid x_i))$, if $A$ is discrete, and $\xi_{x_i, a_i} = 0$, if $A$ is continuous.

**Learning algorithm for stage 2:** There are two remaining challenges we need to address in Stage 2: (i) optimizing Eq. (6) does not ensure that the sensitivity constraints imposed by the GTSM $\mathcal{M}$ hold; and (ii) computing the Stage 2 loss from Eq. (6) may not be analytically tractable. For (i), we propose to incorporate the sensitivity constraints by using the augmented Lagrangian method (Nocedal & Wright, 2006), which has already been successfully applied in the context of partial identification with neural networks (Padh et al., 2023; Schröder et al., 2024). For (ii), we propose to obtain samples $\widetilde{u} = (\widetilde{u}^{(j)}_{x,a})_{j=1}^k \overset{\text{i.i.d.}}{\sim} \mathcal{N}(0_{d_y}, I_{d_y})$ and $\xi = (\xi^{(j)}_{x,a})_{j=1}^k \overset{\text{i.i.d.}}{\sim} \text{Bernoulli}(\mathbb{P}_{\text{obs}}(a \mid x))$

together with Monte Carlo estimators $\hat{\mathcal{L}}_2(\eta, \widetilde{u}, \xi)$ of the Stage 2 loss $\mathcal{L}_2(\eta)$ and $\hat{\mathcal{D}}_{x,a}(\eta, \widetilde{u})$ of the sensitivity constraint $\mathcal{D}_{x,a}(\mathbb{P}(U \mid x), \mathbb{P}(U \mid x, a))$. We refer to Appendix E for details, including instantiations of our framework for numerous sensitivity models and causal queries.

**Implementation:** We use autoregressive neural spline flows (Durkan et al., 2019; Dolatabadi et al., 2020). For estimating propensity scores $\mathbb{P}_{\text{obs}}(a \mid x)$, we use fully connected neural networks with softmax activation. We perform training using the Adam optimizer (Kingma & Ba, 2015). We choose the number of epochs such that NEURALCSA satisfies the sensitivity constraint for a given sensitivity parameter. Details are in Appendix F.

## 6 EXPERIMENTS

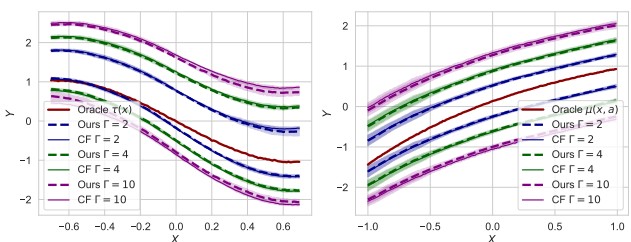

We now demonstrate the effectiveness of NEURALCSA for causal sensitivity analysis empirically. As is common in the causal inference literature, we use synthetic and semi-synthetic data with known causal ground truth to evaluate NEURALCSA (Kallus et al., 2019; Jesson et al., 2022). We proceed as follows: (i) We use synthetic data to show the validity of bounds from

Figure 5: Validating the correctness of NEURALCSA (ours) by comparing with optimal closed-form solutions (CF) for the MSM on simulated data. *Left:* Dataset 1, binary treatment. *Right:* Dataset 2, continuous treatment. Reported: mean ± standard deviation over 5 runs.

NEURALCSA under multiple sensitivity models, treatment types, and causal queries. We also show that for the MSM, the NEURALCSA bounds coincide with known optimal solutions. (ii) We show the validity of the NEURALCSA bounds using a semi-synthetic dataset. (iii) We show the applicability of NEURALCSA in a case study using a real-world dataset with multiple outcomes, which cannot be handled by previous approaches. We refer to Appendix D for details regarding datasets and experimental evaluation, and to Appendix H for additional experiments.

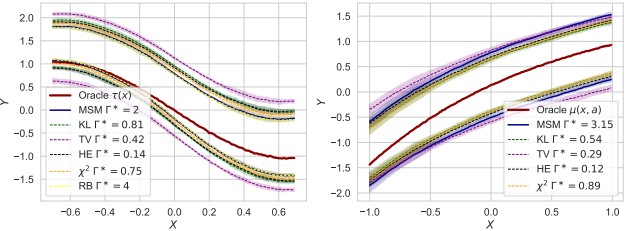

Figure 6: Confirming the validity of our NEURALCSA bounds for various sensitivity models. *Left:* Dataset 1, binary treatment. *Right:* Dataset 2, continuous treatment. Reported: mean ± standard deviation over 5 runs.

**(i) Synthetic data:** We consider two synthetic datasets of sample size $n = 10000$ inspired from previous work on sensitivity analysis: Dataset 1 is adapted from Kallus et al. (2019) and has a binary treatment $A \in \{0, 1\}$. The data-generating process follows an MSM with oracle sensitivity parameter $\Gamma^* = 2$. We are interested in the CATE $\tau(x) = \mathbb{E}[Y(1) - Y(0) \mid x]$. Dataset 2 is adapted from Jesson et al. (2022) and has a continuous treatment $A \in [0, 1]$. Here, we are interested in the dose-response function $\mu(x, a) = \mathbb{E}[Y(a) \mid x]$, where we choose $a = 0.5$. We report results for further treatment values in Appendix H.

We first compare our NEURALCSA bounds with existing results closed-form bounds (CF) for the MSM (Dorn & Guo, 2022; Frauen et al., 2023b), which have been proven to be optimal. We plot both NEURALCSA and the CF for both datasets and three choices of sensitivity parameter $\Gamma \in \{2, 4, 10\}$ (Fig. 5). Our bounds almost coincide with the optimal CF solutions, which confirms that NEURALCSA learns optimal bounds under the MSM.

We also show the validity of our NEURALCSA bounds for Rosenbaum's sensitivity model and the following $f$-sensitivity models: Kullbach-Leibler (KL, $f(x) = x \log(x)$), Total Variation (TV, $f(x) = 0.5|x - 1|$), Hellinger (HE, $f(x) = (\sqrt{x} - 1)^2$), and Chi-squared ($\chi^2$, $f(x) = (x - 1)^2$). To do so, we choose the ground-truth sensitivity parameter $\Gamma^*$ for each sensitivity model that satisfies the respective sensitivity constraint (see Appendix G for details). The results are in Fig. 6. We make the following observations: (i) all bounds cover the causal query on both datasets, thus confirming the validity of NEURALCSA. (ii) For Dataset 1, the MSM returns the tightest bounds because our simulation follows an MSM.

**(ii) Semi-synthetic data:** We create a semi-synthetic dataset using MIMIC-III (Johnson et al., 2016), which includes electronic health records from patients admitted to intensive care units. We extract 8 confounders and a binary treatment (mechanical ventilation). Then, we augment the data with a synthetic unobserved confounder and outcome. We obtain $n = 14719$ patients and split the data into train (80%), val (10%), and test (10%). For details, see Appendix G.

We verify the validity of our NEURALCSA bounds for CATE in the following way: For each sensitivity model, we obtain the smallest oracle sensitivity parameter $\Gamma^*$ that guarantees coverage (i.e., satisfies the respective sensitivity constraint) for 50% of the test samples. Then, we plot the coverage and median interval length of the NEURALCSA bounds over the test set. The results are in Table 2.

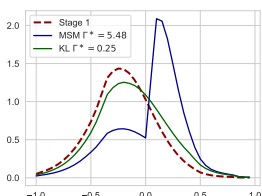

Figure 7: Analytic stage 2 densities for MSM and KL-sensitivity model (upper bounds).

We observe that (i) all bounds achieve at least 50% coverage, thus confirming the validity of the bounds, and (ii) some sensitivity models (e.g., the MSM) are conservative, i.e., achieve much higher coverage and interval length than needed. This is because the sensitivity constraints of these models do not adapt well to the data-generating process, thus the need for choosing a large $\Gamma^*$ to guarantee coverage. This highlights the importance of choosing a sensitivity model that captures the data-generating process well. For further details, we refer to (Jin et al., 2022). We also provide further insights into the difference between two exemplary sensitivity models: the MSM and the KL-sensitivity model. To do so, we plot the observational distribution from stage 1 together with the shifted distributions from stage 2 that lead to the respective upper bound for a fixed test patient (Fig. 7). The distribution shift corresponding to the MSM is a step function, which is consistent with results from established literature (Jin et al., 2023). This is in contrast to the smooth distribution shift obtained by the KL-sensitivity model. In addition, this example illustrates the possibility of using NEURALCSA for sensitivity analysis on the *entire interventional density*.

**(iii) Case study using real-world data:** We now demonstrate an application of NEURALCSA to perform causal sensitivity analysis for an interventional distribution on multiple outcomes. To do so, we use the same MIMIC-III data from our semi-synthetic experiments but add two outcomes: heart rate ($Y_1$) and blood pressure ($Y_2$). We consider the causal query $\mathbb{P}(Y_1(1) \geq 115, Y_2(1) \geq 90 \mid X = x)$, i.e., the joint probability of achieving a heart rate higher than 115 and a blood pressure higher than 90

Table 2: Results for semi-synthetic data

| Sensitivity model | Coverage | Interval length |
|---|---|---|
| MSM $\Gamma^* = 5.48$ | $0.91 \pm 0.03$ | $0.77 \pm 0.03$ |
| KL $\Gamma^* = 0.25$ | $0.54 \pm 0.07$ | $0.31 \pm 0.01$ |
| TV $\Gamma^* = 0.38$ | $0.86 \pm 0.09$ | $0.83 \pm 0.14$ |
| HE $\Gamma^* = 0.18$ | $0.83 \pm 0.06$ | $0.63 \pm 0.03$ |
| $\chi^2$ $\Gamma^* = 0.68$ | $0.67 \pm 0.07$ | $0.41 \pm 0.01$ |
| RB $\Gamma^* = 14.42$ | $0.79 \pm 0.07$ | $0.56 \pm 0.03$ |

Reported: mean $\pm$ standard deviation (5 runs).

under treatment intervention ("danger area"). We consider an MSM and train NEURALCSA with sensitivity parameters $\Gamma \in \{2, 4\}$. Then, we plot the stage 1 distribution together with both stage 2 distributions for a fixed, untreated patient from the test set in Fig. 8.

As expected, increasing $\Gamma$ leads to a distribution shift in the direction of the "danger area", i.e., high heart rate and high blood pressure. For $\Gamma = 2$, there is only a moderate fraction of probability mass inside the danger area, while, for $\Gamma = 4$, this fraction is much larger. A practitioner may potentially decide against treatment if there are other unknown factors (e.g., undetected comorbidity) that could result in a confounding strength of $\Gamma = 4$.

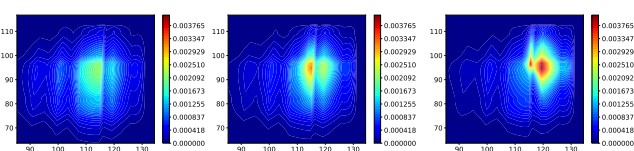

Figure 8: Contour plots of 2D densities obtained by NEURALCSA under an MSM. Here, we aim to learn an upper bound of the causal query $\mathbb{P}(Y_1(1) \geq 115, Y_2(1) \geq 90 \mid X = x_0)$ for a test patient $x_0$. *Left:* Stage 1/ observational distribution. *Middle:* Stage 2, $\Gamma = 2$. *Right:* Stage 2, $\Gamma = 4$.

**Conclusion.** From a methodological perspective, NEURALCSA offers new ideas to causal sensitivity analysis and partial identification: In contrast to previous methods, NEURALCSA explicitly learns a latent distribution shift due to treatment intervention. We refer to Appendix I for a discussion on limitations and future work. From an applied perspective, NEURALCSA enables practitioners to perform causal sensitivity analysis in numerous settings, including multiple outcomes. Furthermore, it allows for choosing from a wide variety of sensitivity models, which may be crucial to effectively incorporate domain knowledge about the data-generating process.

**Acknowledgements.** S.F. acknowledges funding via Swiss National Science Foundation Grant 186932.

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

## A  EXTENDED RELATED WORK

In the following, we provide an extended related work. Specifically, we elaborate on (1) a systematic overview of causal sensitivity analysis, (2) its application in settings beyond partial identification of interventional causal queries, and (3) point identification and estimation of causal queries.

### A.1  A SYSTEMATIC OVERVIEW ON CAUSAL SENSITIVITY ANALYSIS

In Table 3, we provide a systematic overview of existing works for causal sensitivity analysis, which we group by the underlying sensitivity model, the treatment type, and the causal query. As such, Table 3 extends Table 1 in that we follow the same categorization but now point to the references from the literature.

Table 3: Overview of key works for causal sensitivity analyses under the MSM, $f$-sensitivity models, or Rosenbaum's sensitivity model. Settings with no existing literature are indicated with a red cross (✗). Treatments are either binary or continuous. NEURALCSA framework is applicable in all settings.

| Sensitivity model
Causal query | MSM | | $f$-sensitivity | | Rosenbaum | |
|---|---|---|---|---|---|---|
| | Binary | Cont.[†] | Binary | Cont. | Binary | Cont. |
| CATE | Tan (2006)
Kallus et al. (2019)
Zhao et al. (2019)
Jesson et al. (2021)
Dorn & Guo (2022)
Dorn et al. (2022)
Oprescu et al. (2023)
Soriano et al. (2023) | Bonvini et al. (2022)
Jesson et al. (2022)
Frauen et al. (2023b) | Jin et al. (2022) | ✗ | Rosenbaum & Rubin (1983b)
Rosenbaum (1987)
Heng & Small (2021) | ✗ |
| Distributional effects | Frauen et al. (2023b) | (Frauen et al., 2023b) | ✗ | ✗ | ✗ | ✗ |
| Interventional density | Jin et al. (2023)
Yin et al. (2022)
Marmarelis et al. (2023b)
Frauen et al. (2023b) | Frauen et al. (2023b) | Jin et al. (2022) | ✗ | ✗ | ✗ |
| Multiple outcomes | ✗ | ✗ | ✗ | ✗ | ✗ | ✗ |

([†]) The MSM for continuous treatment is also called continuous MSM (CMSM) (Jesson et al., 2022).

Evidently, many works have focused on sensitivity analysis for CATE in binary treatment settings. For many settings, such as $f$-sensitiivity and Rosenbaum's sensitivity model with continuous treatments or multiple outcomes, no previous work exists. Here, NEURALCSA is the first work that allows for computing bounds in these settings.

### A.2  SENSITIVITY ANALYSIS IN OTHER CAUSAL SETTINGS

Causal sensitivity analysis has found applicability not only in addressing the partial identification problem, as discussed in Eq. (1), but also in various domains of machine learning and causal inference. We briefly highlight some notable instances where ideas from causal sensitivity analysis have made substantial contributions.

One such stream of literature is off-policy learning, where sensitivity models have been leveraged to account for unobserved confounding or distribution shifts (Kallus & Zhou, 2018; Hatt et al., 2022). Here, sensitivity analysis enables robust policy learning. Another example is algorithmic fairness, where sensitivity analysis has been used to study causal fairness notions (e.g., counterfactual fairness) under unobserved confounding (Kilbertus et al., 2019). Finally have been used to study the partial identification of counterfactual queries (Melnychuk et al., 2023b)

### A.3  POINT IDENTIFICATION AND ESTIMATION

If we replace the latent unconfoundedness assumption in Assumtion 1 with (non-latent) unconfoundedness, that is,

$$Y(a) \perp\!\!\!\perp A \mid X \quad \text{for all} \quad a \in \mathcal{A}, \tag{7}$$

we can point-identify the distribution of the potential outcomes via

$$\mathbb{P}(Y(a) = y \mid x) = \mathbb{P}(Y(a) = y \mid x, a) = \mathbb{P}_{\text{obs}}(y \mid x, a). \tag{8}$$

Hence, inferring the causal query $Q(x, a, \mathbb{P})$ reduces to a purely statistical inference problem, i.e., estimating $\mathcal{F}(\mathbb{P}_{\mathrm{obs}}(Y \mid x, a))$ from finite data.

Various methods for estimating point-identified causal effects under unconfoundedness have been proposed. In recent years, a particular emphasis has been on methods for estimating (conditional) average treatment effects (CATEs) that make use of machine learning to model flexible non-linear relationships within the data. Examples include forest-based methods (Wager & Athey, 2018) and deep learning (Johansson et al., 2016; Shalit et al., 2017; Yoon et al., 2018; Shi et al., 2019). Another stream of literature incorporates theory from semi-parametric statistics and provides robustness and efficiency guarantees (van der Laan & Rubin, 2006; Chernozhukov et al., 2018; Künzel et al., 2019; Curth & van der Schaar, 2021; Kennedy, 2023). Beyond CATE, methods have also been proposed for estimating distributional effects or potential outcome densities (Chernozhukov et al., 2013; Muandet et al., 2021; Kennedy et al., 2023). In particular, Melnychuk et al. (2023a) proposed normalizing flows for potential outcome densities. Finally, Schweisthal et al. (2023) leveraged normalizing flows for estimating the generalized propensity score in a setting with continuous treatment. We emphasize that all these methods focus on *estimation* of *point-identified* causal queries, while we are interested in *causal sensitivity analysis* and thus *partial identification* under violations of the unconfoundedness assumption.

## B PROOFS

### B.1 PROOF OF LEMMA 1

We provide a proof for the following more detailed version of Lemma 1.

**Lemma 3.** *The MSM, the $f$-sensitivity model, and Rosenbaum's sensitivity model are GTSMs with sensitivity parameter $\Gamma$. Let $\rho(x, u, a) = \frac{1}{1-\mathbb{P}(a|x)} \left( \frac{\mathbb{P}(u|x)}{\mathbb{P}(u|x,a)} - \mathbb{P}(a \mid x) \right)$ and $\rho(x, u_1, u_2, a) = \frac{\mathbb{P}(u_1|x,a)\mathbb{P}(u_2|x)-\mathbb{P}(u_1|x,a)\mathbb{P}(u_2|x,a)\mathbb{P}(a|x)}{\mathbb{P}(u_2|x,a)\mathbb{P}(u_1|x)-\mathbb{P}(u_1|x,a)\mathbb{P}(u_2|x,a)\mathbb{P}(a|x)}$. For the MSM, we have*

$$\mathcal{D}_{x,a}(\mathbb{P}(U \mid x), \mathbb{P}(U \mid x, a)) = \max \left\{ \sup_{u \in \mathcal{U}} \rho(x, u, a), \sup_{u \in \mathcal{U}} \rho(x, u, a)^{-1} \right\}. \tag{9}$$

*For $f$-sensitivity models, we have*

$$\mathcal{D}_{x,a}(\mathbb{P}(U \mid x), \mathbb{P}(U \mid x, a)) = \max \left\{ \int_{\mathcal{U}} f(\rho(x, u, a))\mathbb{P}(u \mid x, a) \, \mathrm{d}u, \int_{\mathcal{U}} f(\rho(x, u, a)^{-1})\mathbb{P}(u \mid x, a) \, \mathrm{d}u \right\}. \tag{10}$$

*For Rosenbaum's sensitivity model, we have*

$$\mathcal{D}_{x,a}(\mathbb{P}(U \mid x), \mathbb{P}(U \mid x, a)) = \max \left\{ \sup_{u_1, u_2 \in \mathcal{U}} \rho(x, u_1, u_2, a), \sup_{u_1, u_2 \in \mathcal{U}} \rho(x, u_1, u_2, a)^{-1} \right\}. \tag{11}$$

*Proof.* We show that all three sensitivity models (MSM, $f$-sensitivity models, and Rosenbaum's sensitivity model) are GTSMs. Recall that the odds ratio is defined as $\mathrm{OR}(a, b) = \frac{a}{(1-a)} \frac{(1-b)}{b}$.

**MSM:** Using Bayes' theorem, we obtain $\mathbb{P}(u \mid x, a) = \frac{\mathbb{P}(a|x,u)\mathbb{P}(u|x)}{\mathbb{P}(a|x)}$ and therefore

$$\rho(x, u, a) = \frac{1}{1 - \mathbb{P}(a \mid x)} \left( \frac{\mathbb{P}(u \mid x)}{\mathbb{P}(u \mid x, a)} - \mathbb{P}(a \mid x) \right) \tag{12}$$

$$= \frac{1}{1 - \mathbb{P}(a \mid x)} \left( \frac{\mathbb{P}(a \mid x)}{\mathbb{P}(a \mid x, u)} - \mathbb{P}(a \mid x) \right) \tag{13}$$

$$= \frac{\mathbb{P}(a \mid x)}{1 - \mathbb{P}(a \mid x)} \left( \frac{1 - \mathbb{P}(a \mid x, u)}{\mathbb{P}(a \mid x, u)} \right) \tag{14}$$

$$= \mathrm{OR}\left(\mathbb{P}(a \mid x), \mathbb{P}(a \mid x, u)\right). \tag{15}$$

Hence, $\max \left\{ \sup_{u \in \mathcal{U}} \rho(x, u, a), \sup_{u \in \mathcal{U}} \rho(x, u, a)^{-1} \right\} \leq \Gamma$ is equivalent to

$$\frac{1}{\Gamma} \leq \mathrm{OR}\left(\mathbb{P}(a \mid x), \mathbb{P}(a \mid x, u)\right) \leq \Gamma \tag{16}$$

for all $u \in \mathcal{U}$, which reduces to the original MSM defintion for $a = 1$.

$f$**-sensitivity models:** Follows immediately from $\rho(x, u, a) = \mathrm{OR}\left(\mathbb{P}(a \mid x), \mathbb{P}(a \mid x, u)\right)$.

**Rosenbaum's sensitivity model:** We can write

$$\rho(x, u_1, u_2, a) = \frac{\mathbb{P}(u_1 \mid x, a)\mathbb{P}(u_2 \mid x, a)\mathbb{P}(a \mid x) - \mathbb{P}(u_1 \mid x, a)\mathbb{P}(u_2 \mid x)}{\mathbb{P}(u_1 \mid x, a)\mathbb{P}(u_2 \mid x, a)\mathbb{P}(a \mid x) - \mathbb{P}(u_2 \mid x, a)\mathbb{P}(u_1 \mid x)} \tag{17}$$

$$= \frac{\mathbb{P}(a \mid x, u_1)\mathbb{P}(u_1 \mid x)}{\mathbb{P}(a \mid x, u_2)\mathbb{P}(u_2 \mid x)} \left( \frac{\mathbb{P}(a \mid x, u_2)\mathbb{P}(u_2 \mid x) - \mathbb{P}(u_2 \mid x)}{\mathbb{P}(a \mid x, u_1)\mathbb{P}(u_1 \mid x) - \mathbb{P}(u_1 \mid x)} \right) \tag{18}$$

$$= \frac{\mathbb{P}(a \mid x, u_1)}{\mathbb{P}(a \mid x, u_2)} \left( \frac{\mathbb{P}(a \mid x, u_2) - 1}{\mathbb{P}(a \mid x, u_1) - 1} \right) \tag{19}$$

$$= \mathrm{OR}\left(\mathbb{P}(a \mid x, u_1), \mathbb{P}(a \mid x, u_2)\right). \tag{20}$$

Hence, $\max \left\{ \sup_{u_1, u_2 \in \mathcal{U}} \rho(x, u_1, u_2, a), \sup_{u_1, u_2 \in \mathcal{U}} \rho(x, u_1, u_2, a)^{-1} \right\} \leq \Gamma$ is equivalent to

$$\frac{1}{\Gamma} \leq \mathrm{OR}\left(\mathbb{P}(a \mid x, u_1), \mathbb{P}(a \mid x, u_2)\right) \leq \Gamma \tag{21}$$

for all $u_1, u_2 \in \mathcal{U}$, which reduces to the original definition of Rosenbaum's sensitivity model for $a = 1$. $\qquad \square$

### B.2 PROOF OF LEMMA 2

*Proof.* We show transformation-invariance separately for all three sensitivity models (MSM, $f$-sensitivity models, and Rosenbaum's sensitivity model).

**MSM:** Let $\mathcal{D}_{x,a}(\mathbb{P}(U \mid x), \mathbb{P}(U \mid x, a)) = \Gamma$, which implies that implies $\frac{1}{\Gamma} \leq \rho(x, u, a) \leq \Gamma$ for all $u \in \mathcal{U}$. By rearranging terms, we obtain

$$\mathbb{P}(u \mid x) \leq (\Gamma(1 - \mathbb{P}(a \mid x)) + \mathbb{P}(a \mid x)) \ \mathbb{P}(u \mid x, a) \tag{22}$$

and

$$\mathbb{P}(u \mid x) \geq \left(\frac{1}{\Gamma}(1 - \mathbb{P}(a \mid x)) + \mathbb{P}(a \mid x)\right) \mathbb{P}(u \mid x, a). \tag{23}$$

Let $t \colon \mathcal{U} \to \widetilde{\mathcal{U}}$ be a transformation of the unobserved confounder. By using Eq. (22), we can write

$$\rho(x, t(u), a) = \frac{1}{1 - \mathbb{P}(a \mid x)} \left( \frac{\mathbb{P}(t(u) \mid x)}{\mathbb{P}(t(u) \mid x, a)} - \mathbb{P}(a \mid x) \right) \tag{24}$$

$$= \frac{1}{1 - \mathbb{P}(a \mid x)} \left( \frac{\int \delta(t(u) - t(u'))\mathbb{P}(u' \mid x)\,\mathrm{d}u'}{\int \delta(t(u) - t(u'))\mathbb{P}(u' \mid x, a)\,\mathrm{d}u'} - \mathbb{P}(a \mid x) \right) \tag{25}$$

$$\leq \frac{1}{1 - \mathbb{P}(a \mid x)} \left( \frac{\int \delta(t(u) - t(u'))\mathbb{P}(u' \mid x, a)\,\mathrm{d}u'}{\int \delta(t(u) - t(u'))\mathbb{P}(u' \mid x, a)\,\mathrm{d}u'} \right) \Gamma \tag{26}$$

$$= \Gamma \tag{27}$$

for all $u \in \mathcal{U}$. Similarly, we can use Eq. (23) to obtain

$$\rho(x, t(u), a) = \frac{1}{1 - \mathbb{P}(a \mid x)} \left( \frac{\int \delta(t(u) - t(u'))\mathbb{P}(u' \mid x)\,\mathrm{d}u'}{\int \delta(t(u) - t(u'))\mathbb{P}(u' \mid x, a)\,\mathrm{d}u'} - \mathbb{P}(a \mid x) \right) \tag{28}$$

$$\geq \frac{1}{1 - \mathbb{P}(a \mid x)} \left( \frac{\int \delta(t(u) - t(u'))\mathbb{P}(u' \mid x, a)\,\mathrm{d}u'}{\int \delta(t(u) - t(u'))\mathbb{P}(u' \mid x, a)\,\mathrm{d}u'} \right) \frac{1}{\Gamma} \tag{29}$$

$$= \frac{1}{\Gamma} \tag{30}$$

for all $u \in \mathcal{U}$. Hence,

$$\mathcal{D}_{x,a}(\mathbb{P}(t(U) \mid x), \mathbb{P}(t(U) \mid x, a)) = \max \left\{ \sup_{u \in \mathcal{U}} \rho(x, t(u), a), \ \sup_{u \in \mathcal{U}} \rho(x, t(u), a)^{-1} \right\} \leq \Gamma. \tag{31}$$

$f$**-sensitivity models:** This follows from the data compression theorem for $f$-divergences. We refer to Polyanskiy & Wu (2022) for details.

**Rosenbaum's sensitivity model**: We begin by rewriting

$$\rho(x, u_1, u_2, a) = \frac{\mathbb{P}(u_1 \mid x, a)\mathbb{P}(u_2 \mid x, a)\mathbb{P}(a \mid x) - \mathbb{P}(u_1 \mid x, a)\mathbb{P}(u_2 \mid x)}{\mathbb{P}(u_1 \mid x, a)\mathbb{P}(u_2 \mid x, a)\mathbb{P}(a \mid x) - \mathbb{P}(u_2 \mid x, a)\mathbb{P}(u_1 \mid x)} \tag{32}$$

$$= \left( \frac{1}{\frac{\mathbb{P}(u_1 \mid x)}{\mathbb{P}(u_1 \mid x, a)} - \mathbb{P}(a \mid x)} \right) \left( \frac{\mathbb{P}(u_2 \mid x)}{\mathbb{P}(u_2 \mid x, a)} - \mathbb{P}(a \mid x) \right) \tag{33}$$

as a function of density ratios on $\mathcal{U}$. Let now $\mathcal{D}_{x,a}(\mathbb{P}(U \mid x), \mathbb{P}(U \mid x, a)) = \Gamma$. This implies

$$\mathbb{P}(u_1 \mid x) \leq \left( \Gamma \left( \frac{\mathbb{P}(u_2 \mid x)}{\mathbb{P}(u_2 \mid x, a)} - \mathbb{P}(a \mid x) \right) + \mathbb{P}(a \mid x) \right) \mathbb{P}(u_1 \mid x, a) \tag{34}$$

and

$$\mathbb{P}(u_1 \mid x) \geq \left( \frac{1}{\Gamma} \left( \frac{\mathbb{P}(u_2 \mid x)}{\mathbb{P}(u_2 \mid x, a)} - \mathbb{P}(a \mid x) \right) + \mathbb{P}(a \mid x) \right) \mathbb{P}(u_1 \mid x, a) \tag{35}$$

for all $u_1, u_2 \in \mathcal{U}$. Let $t \colon \mathcal{U} \to \widetilde{\mathcal{U}}$ be a transformation. By using Eq. (34) and Eq. (35), we obtain

$$\rho(x, t(u_1), t(u_1), a) = \left( \frac{1}{\frac{\int \delta(t(u_1) - t(u_1'))\mathbb{P}(u_1' \mid x)\, \mathrm{d}u_1'}{\int \delta(t(u_1) - t(u_1'))\mathbb{P}(u_1' \mid x, a)\, \mathrm{d}u_1'} - \mathbb{P}(a \mid x)} \right) \tag{36}$$

$$\left( \frac{\int \delta(t(u_2) - t(u_2'))\mathbb{P}(u_2' \mid x)\, \mathrm{d}u_2'}{\int \delta(t(u_2) - t(u_2'))\mathbb{P}(u_2' \mid x, a)\, \mathrm{d}u_2'} - \mathbb{P}(a \mid x) \right) \tag{37}$$

$$\leq \left( \frac{1}{\frac{1}{\Gamma}\left( \frac{\mathbb{P}(u_2 \mid x)}{\mathbb{P}(u_2 \mid x, a)} - \mathbb{P}(a \mid x) \right)} \right) \Gamma \left( \frac{\mathbb{P}(u_1 \mid x)}{\mathbb{P}(u_1 \mid x, a)} - \mathbb{P}(a \mid x) \right) \tag{38}$$

$$= \frac{\Gamma^2}{\rho(x, u_1, u_1, a)} \tag{39}$$

for all $u_1, u_2 \in \mathcal{U}$. Hence,

$$\inf_{u_1, u_2} \rho(x, t(u_1), t(u_1), a) \leq \Gamma. \tag{40}$$

By using analogous arguments, we can also show that

$$\sup_{u_1, u_2} \rho(x, t(u_1), t(u_1), a) \leq \Gamma, \tag{41}$$

which implies

$$\mathcal{D}_{x,a}\big(\mathbb{P}(t(U) \mid x), \mathbb{P}(t(U) \mid x, a)\big) \leq \Gamma. \tag{42}$$

$$\square$$

### B.3 PROOF OF THEOREM 1

Before stating the formal proof for Theorem 1, we provide a sketch to give an overview of the main ideas and intuition.

*Why is it sufficient to only consider invertible functions $f_{x,a}^* \colon \mathcal{U} \to \mathcal{Y}$, i.e., model $\mathbb{P}^*(Y \mid x, a, u) = \delta(Y - f_{x,a}^*(u))$ as a Dirac delta distribution?* Here, it is helpful to have a closer look at Fig. 1: Intervening on the treatment $A$ causes a shift in the latent distribution of $U$, which then leads to a shifted interventional distribution $\mathbb{P}(Y(a) = y \mid x)$. The question is now: How can we obtain an interventional distribution that results in a maximal causal query (for the upper bound)? Let us consider a structural equation of the form $Y = g_{x,a}^*(U, \epsilon)$, where $\epsilon$ is some independent noise. Hence, the "randomness" (entropy) in $Y$ comes from both $U$ and $\epsilon$, however, the distribution shift only arises through $U$. Intuitively, the interventional distribution should be maximally shifted if $Y$ only depends on the unobserved confounder and not on independent noise, i.e., $g_{x,a}^*(U, \epsilon) = f_{x,a}^*(U)$ for some invertible function $f_{x,a}^*$. One may also think about this as achieving the maximal "dependence" (mutual information) between the random variables $U$ and $Y$. Note that any GTSM only restricts the dependence between $U$ and $A$, but not between $U$ and $Y$.

*Why can we fix $\mathbb{P}^*(U \mid x, a)$ and $f_{x,a}^*$ without losing the ability to achieve the optimum in Eq.* (4). The basic idea is as follows: Let $\widetilde{\mathbb{P}}(\widetilde{U} \mid x, a)$ and $\widetilde{f}x, a$ be optimal solutions to Eq. (4) for a potentially different latent variable $\widetilde{U}$. Then, we can define a mapping $t = f_{x,a}^{*-1} \circ \widetilde{f}_{x,a} \colon \widetilde{U} \to U$ between latent spaces that transforms $\widetilde{\mathbb{P}}(\widetilde{U} \mid x, a)$ into our fixed $\mathbb{P}^*(U \mid x, a)$ (because both $\widetilde{f}_{x,a}$ and $f_{x,a}^*$ respect the observational distribution). Furthermore, we can use $t$ to push the optimal shifted distribution $\widetilde{\mathbb{P}}(\widetilde{U} \mid x)$ (under treatment intervention) to the latent variable $U$ (see Eq. (46)). We will show that this is sufficient to obtain a distribution $\mathbb{P}^*$ that induces $\mathbb{P}^*(U \mid x, a)$ and $\mathbb{P}^*(Y \mid x, u, a) = \delta(Y - f_{x,a}^*(u))$ and which satisfies the sensitivity constraints. For the latter property, we require the sensitivity model to be "invariant" with respect to the transformation $t$, for which we require our transformation-invariance assumption (Definition 3).

We proceed now with our formal proof of Theorem 1.

*Proof.* Without loss of generality, we provide a proof for the upper bound $Q_{\mathcal{M}}^+(x, a)$. Our arguments work analogously for the lower bound $Q_{\mathcal{M}}^-(x, a)$. Furthermore, we only show the inequality

$$Q_{\mathcal{M}}^+(x, a) \leq \sup_{\mathbb{P}^* \in \mathcal{P}^*} Q(x, a, \mathbb{P}^*), \tag{43}$$

because the other direction ("$\geq$") holds by definition of $Q_{\mathcal{M}}^+(x,a)$. Hence, it is enough to show the existence of a sequence of full distributions $(\mathbb{P}_\ell^*)_{\ell \in \mathbb{N}}$ with $\mathbb{P}_\ell^* \in \mathcal{P}^*$ for all $\ell \in \mathbb{N}$ that satisfies $\lim_{\ell \to \infty} Q(x,a,\mathbb{P}_\ell^*) = Q_{\mathcal{M}}^+(x,a)$.

To do so, we proceed in three steps: In step 1, we construct a sequence $(\mathbb{P}_\ell^*)_{\ell \in \mathbb{N}}$ of full distributions that induce $\mathbb{P}_\ell^*(U \mid x,a) = \mathbb{P}^*(U \mid x,a)$ and $\mathbb{P}_\ell^*(Y \mid x,u,a) = \mathbb{P}^*(Y \mid x,u,a) = \delta(Y - f_{x,a}^*(u))$ for every $\ell \in \mathbb{N}$. In step 2, we show compatibility with the sensitivity model, i.e., $\mathbb{P}_\ell^* \in \mathcal{M}$ for all $\ell \in \mathbb{N}$. Finally, in step 3, we show that $\lim_{\ell \to \infty} Q(x,a,\mathbb{P}_\ell^*) = Q_{\mathcal{M}}^+(x,a)$.

**Step 1:** Let $\widetilde{\mathbb{P}}$ be a full distribution on $\mathcal{X} \times \widetilde{\mathcal{U}} \times \mathcal{A} \times \mathcal{Y}$ for some latent space $\widetilde{\mathcal{U}}$ that is the solution to Eq. (1). By definition, there exists a sequence $(\widetilde{\mathbb{P}}_\ell)_{\ell \in \mathbb{N}}$ with $\widetilde{\mathbb{P}}_\ell \in \mathcal{M}$ and $\lim_{\ell \to \infty} Q(x,a,\widetilde{\mathbb{P}}_\ell) = Q_{\mathcal{M}}^+(x,a)$. Let $\widetilde{\mathbb{P}}_\ell(\widetilde{U} \mid x)$ and $\widetilde{\mathbb{P}}_\ell(Y \mid x,u,a)$ be corresponding induced distributions (for fixed $x$, $a$). Without loss of generality, we can assume that $\widetilde{\mathbb{P}}_\ell$ is induced by a structural causal model (Pearl, 2009), so that we can write the conditional outcome distribution with a (not necessarily invertible) functional assignment $Y = \widetilde{f}_{X,A,\ell}(U)$ as a point distribution $\widetilde{\mathbb{P}}_\ell(Y \mid x,u,a) = \delta(Y - \widetilde{f}_{x,a,l}(u))$. Note that we do not explicitly consider exogenous noise because we can always consider this part of the latent space $\widetilde{\mathcal{U}}$. By Eq. (3) and Eq. (2) we can write the observed conditional outcome distribution as

$$\mathbb{P}_{\mathrm{obs}}(Y \mid x,a) = \widetilde{\mathbb{P}}_\ell(\widetilde{f}_{x,a,\ell}(\widetilde{U}) \mid x,a), \tag{44}$$

and the potential outcome distribution conditioned on $x$ as

$$\widetilde{\mathbb{P}}_\ell(Y(a) \mid x) = \widetilde{\mathbb{P}}_\ell(\widetilde{f}_{x,a,\ell}(\widetilde{U}) \mid x). \tag{45}$$

We now define the sequence $(\mathbb{P}_\ell^*)_{\ell \in \mathbb{N}}$. First we define a distribution on $\mathcal{U} \subseteq \mathbb{R}^{d_y}$ via

$$\mathbb{P}_\ell^*(U \mid x) = \widetilde{\mathbb{P}}_\ell(f_{x,a}^{*\,-1}(Y(a)) \mid x) = \widetilde{\mathbb{P}}_\ell(f_{x,a}^{*\,-1}\left(\widetilde{f}_{x,a,\ell}(\widetilde{U})\right) \mid x). \tag{46}$$

We then define full probability distribution $\mathbb{P}_\ell^*$ for the fixed $x$ and $a$ and all $u \in \mathcal{U}, y \in \mathcal{Y}$ as

$$\mathbb{P}_\ell^*(x,u,a,y) = \delta\left(f_{x,a}^*(u) - y\right) \mathbb{P}^*(u \mid x,a) \mathbb{P}_{\mathrm{obs}}(x,a). \tag{47}$$

Finally, we can choose a family of distributions $(\mathbb{P}_\ell^*(U \mid x,a'))_{a' \neq a}$ so that $\mathbb{P}_\ell^*(U \mid x) = \int \mathbb{P}_\ell^*(U \mid x,a) \mathbb{P}_{\mathrm{obs}}(a \mid x)\, \mathrm{d}a$ and define

$$\mathbb{P}_\ell^*(x,u,a',y) = \delta\left(f_{x,a'}^*(u) - y\right) \mathbb{P}_\ell^*(u \mid x,a') \mathbb{P}_{\mathrm{obs}}(x,a'). \tag{48}$$

By definition, $\mathbb{P}_\ell^*$ induces the fixed components $\mathbb{P}^*(U \mid x,a)$ and $\mathbb{P}^*(Y \mid x,u,a) = \delta(Y - f_{x,a}^*(u))$, as well as $\mathbb{P}_\ell^*(U \mid x)$ from Eq. (46) and the observational data distribution $\mathbb{P}_{\mathrm{obs}}(X,A,Y)$.

**Step 2:** We now show that $\mathbb{P}_\ell^*$ respects the sensitivity constraints, i.e., satisfies $\mathbb{P}_\ell^* \in \mathcal{M}$. It holds that

$$\mathbb{P}_\ell^*(U \mid x,a) = \mathbb{P}_\ell^*(f_{x,a}^{*\,-1}\left(f_{x,a}^*(U)\right) \mid x,a) \tag{49}$$

$$\overset{(1)}{=} \mathbb{P}_{\mathrm{obs}}(f_{x,a}^{*\,-1}(Y) \mid x,a) \tag{50}$$

$$\overset{(2)}{=} \widetilde{\mathbb{P}}_\ell(f_{x,a}^{*\,-1}\left(\widetilde{f}_{x,a,\ell}(\widetilde{U})\right) \mid x,a), \tag{51}$$

where (1) holds due to the data-compatibility assumption on $f_{x,a}^*$ and (2) holds due to Eq. (44).

We now define a transformation $t \colon \widetilde{\mathcal{U}} \to \mathcal{U}$ via $t = f_{x,a}^{*\,-1} \circ \widetilde{f}_{x,a,\ell}$. We obtain

$$\mathcal{D}_{x,a}(\mathbb{P}_\ell^*(U \mid x), \mathbb{P}_\ell^*(U \mid x,a)) \overset{(1)}{=} \mathcal{D}_{x,a}(\widetilde{\mathbb{P}}_\ell(t(\widetilde{U}) \mid x), \widetilde{\mathbb{P}}_\ell^*(t(\widetilde{U}) \mid x,a)) \tag{52}$$

$$\overset{(2)}{\leq} \mathcal{D}_{x,a}(\widetilde{\mathbb{P}}_\ell(\widetilde{U} \mid x), \widetilde{\mathbb{P}}_\ell^*(\widetilde{U} \mid x,a)) \tag{53}$$

$$\overset{(3)}{\leq} \Gamma, \tag{54}$$

where (1) holds due to Eq. (46) and Eq. (49), (2) holds due to the tranformation-invariance property of $\mathcal{M}$, and (3) holds because $\widetilde{\mathbb{P}}_\ell \in \mathcal{M}$. Hence, $\mathbb{P}_\ell^* \in \mathcal{M}$.

**Step 3:** We show now that $\lim_{\ell \to \infty} Q(x, a, \mathbb{P}_\ell^*) = Q_{\mathcal{M}}^+(x, a)$, which completes our proof. By Eq. (46), it holds that

$$\mathbb{P}_\ell^*(Y(a) \mid x) = \mathbb{P}_\ell^*(f_{x,a}^*(U) \mid x) = \widetilde{\mathbb{P}}_\ell(Y(a) \mid x), \tag{55}$$

which means that potential outcome distributions conditioned on $x$ coincide for $\mathbb{P}_\ell^*$ and $\widetilde{\mathbb{P}}_\ell$. It follows that

$$Q(x, a, \mathbb{P}_\ell^*) = \mathcal{F}\left(\mathbb{P}_\ell^*(Y(a) \mid x)\right) = \mathcal{F}\left(\widetilde{\mathbb{P}}_\ell(Y(a) \mid x)\right) \xrightarrow[\ell \to \infty]{} Q_{\mathcal{M}}^+(x, a). \tag{56}$$

$\square$

## C  FURTHER SENSITIVITY MODELS

In the following, we list additional sensitivity models that can be written as GTSMs and thus can be used with NEURALCSA.

**Continuous marginal sensitivity model (CMSM)**: The CMSM has been proposed by Jesson et al. (2022) and Bonvini et al. (2022). It is defined via

$$\frac{1}{\Gamma} \le \frac{\mathbb{P}(a \mid x)}{\mathbb{P}(a \mid x, u)} \le \Gamma \tag{57}$$

for all $x \in \mathcal{X}$, $u \in \mathcal{U}$, and $a \in \mathcal{A}$. The CMSM can be written as a CMSM with sensitivity parameter $\Gamma$ by defining

$$\mathcal{D}_{x,a}(\mathbb{P}(u \mid x), \mathbb{P}(u \mid x, a)) = \max \left\{ \sup_{u \in \mathcal{U}} \frac{\mathbb{P}(u \mid x)}{\mathbb{P}(u \mid x, a)}, \sup_{u \in \mathcal{U}} \frac{\mathbb{P}(u \mid x, a)}{\mathbb{P}(u \mid x)} \right\}. \tag{58}$$

This directly follows by applying Bayes' theorem to $\mathbb{P}(u \mid x, a) = \frac{\mathbb{P}(a \mid x, u)\mathbb{P}(u \mid x)}{\mathbb{P}(a \mid x)}$.

**Continuous $f$-sensitivity models**: Motivated by the CMSM, we can define $f$-sensitivity models for continuous treatments via

$$\max \left\{ \int_{\mathcal{U}} f\left( \frac{\mathbb{P}(a \mid x)}{\mathbb{P}(a \mid x, u)} \right) \mathbb{P}(u \mid x, a)\,\mathrm{d}u, \int_{\mathcal{U}} f\left( \frac{\mathbb{P}(a \mid x, u)}{\mathbb{P}(a \mid x)} \right) \mathbb{P}(u \mid x, a)\,\mathrm{d}u \right\} \le \Gamma \tag{59}$$

for all $x \in \mathcal{X}$ and $a \in \mathcal{A}$. By using Bayes' theorem, we can write any continuous $f$-sensitivity model as a GTSM by defining

$$\mathcal{D}_{x,a}(\mathbb{P}(u \mid x), \mathbb{P}(u \mid x, a)) = \max \left\{ \int_{\mathcal{U}} f\left( \frac{\mathbb{P}(u \mid x)}{\mathbb{P}(u \mid x, a)} \right) \mathbb{P}(u \mid x, a)\,\mathrm{d}u, \right. \tag{60}$$

$$\left. \int_{\mathcal{U}} f\left( \frac{\mathbb{P}(u \mid x, a)}{\mathbb{P}(u \mid x)} \right) \mathbb{P}(u \mid x, a)\,\mathrm{d}u \right\}. \tag{61}$$

**Weighted marginal sensitivity models:** Frauen et al. (2023b) proposed a weighted version of the MSM, defined via

$$\frac{1}{(1 - \Gamma)q(x, a) + \Gamma} \le \frac{\mathbb{P}(u \mid x, a)}{\mathbb{P}(u \mid x)} \le \frac{1}{(1 - \Gamma^{-1})q(x, a) + \Gamma^{-1}}, \tag{62}$$

where $q(x, a)$ is a weighting function that incorporates domain knowledge about the strength of unobserved confounding. By using similar arguments as in the proof of Lemma 1, we can write the weighted MSM as a GTSM by defining

$$\mathcal{D}_{x,a}(\mathbb{P}(U \mid x), \mathbb{P}(U \mid x, a)) = \max \left\{ \sup_{u \in \mathcal{U}} \rho(x, u, a), \sup_{u \in \mathcal{U}} \rho(x, u, a)^{-1} \right\}, \tag{63}$$

where

$$\rho(x, u, a) = \frac{1}{1 - q(x, a)} \left( \frac{\mathbb{P}(u \mid x)}{\mathbb{P}(u \mid x, a)} - q(x, a) \right). \tag{64}$$

# D  QUERY AVERAGES AND DIFFERENCES

Here, we show that we can use our bounds $Q_{\mathcal{M}}^{+}(x, a)$ and $Q_{\mathcal{M}}^{-}(x, a)$ to obtain sharp bounds for averages and differences of causal queries. We follow established literature on causal sensitivity analysis (Dorn & Guo, 2022; Dorn et al., 2022; Frauen et al., 2023b).

**Averages:** We are interested in the sharp upper bound for the *average causal query*

$$\bar{Q}_{\mathcal{M}}(a, \mathbb{P}) = \int_{\mathcal{X}} Q(x, a, \mathbb{P}) \, \mathbb{P}_{\text{obs}}(x) \, \mathrm{d}x. \tag{65}$$

An example is the average potential outcome $\mathbb{E}[Y(a)]$, which can be obtained by averaging conditional potential outcomes via $\mathbb{E}[Y(a)] = \int \mathbb{E}(Y(a) \mid x) \, \mathbb{P}_{\text{obs}}(x) \, \mathrm{d}x$. We can obtain upper bounds via

$$\bar{Q}_{\mathcal{M}}^{+}(a) = \sup_{\mathbb{P} \in \mathcal{M}} \int_{\mathcal{X}} Q(x, a, \mathbb{P}) \, \mathbb{P}_{\text{obs}}(x) \, \mathrm{d}x = \int_{\mathcal{X}} Q_{\mathcal{M}}^{+}(x, a) \, \mathbb{P}_{\text{obs}}(x) \, \mathrm{d}x, \tag{66}$$

and lower bounds via

$$\bar{Q}_{\mathcal{M}}^{-}(a) = \inf_{\mathbb{P} \in \mathcal{M}} \int_{\mathcal{X}} Q(x, a, \mathbb{P}) \, \mathbb{P}_{\text{obs}}(x) \, \mathrm{d}x = \int_{\mathcal{X}} Q_{\mathcal{M}}^{-}(x, a) \, \mathbb{P}_{\text{obs}}(x) \, \mathrm{d}x, \tag{67}$$

whenever we can interchange the supremum/ infimum and the integral. That is, bounding the averaged causal query $\bar{Q}_{\mathcal{M}}(a, \mathbb{P})$ reduces to averaging the bounds $Q_{\mathcal{M}}^{+}(x, a)$ and $Q_{\mathcal{M}}^{-}(x, a)$.

**Differences:** For two different treatment values $a_1, a_2 \in \mathcal{A}$, we are interested in the *difference of causal queries*

$$Q(x, a_1, \mathbb{P}) - Q(x, a_2, \mathbb{P}). \tag{68}$$

An example is the conditional average treatment effect $\mathbb{E}[Y(1) \mid x] - \mathbb{E}[Y(0) \mid x]$. We can obtain an upper bound via

$$Q_{\mathcal{M}}^{+}(x, a_1, a_2) = \sup_{\mathbb{P} \in \mathcal{M}} \left( Q(x, a_1, \mathbb{P}) - Q(x, a_2, \mathbb{P}) \right) \tag{69}$$

$$\leq \sup_{\mathbb{P} \in \mathcal{M}} Q(x, a_1, \mathbb{P}) - \inf_{\mathbb{P} \in \mathcal{M}} Q(x, a_2, \mathbb{P}) \tag{70}$$

$$= Q_{\mathcal{M}}^{+}(x, a_1) - Q_{\mathcal{M}}^{-}(x, a_2). \tag{71}$$

Similarly, a lower bound is given by

$$Q_{\mathcal{M}}^{-}(x, a_1, a_2) \geq Q_{\mathcal{M}}^{-}(x, a_1) - Q_{\mathcal{M}}^{+}(x, a_2). \tag{72}$$

It has been shown that these bounds are even sharp for some sensitivity models such as the MSM, i.e., attain equality (Dorn & Guo, 2022).

# E   TRAINING DETAILS FOR NEURALCSA

In this section, we provide details regarding the training of NEURALCSA, in particular, the Monte Carlo estimates for Stage 2 and the full learning algorithm.

## E.1   MONTE CARLO ESTIMATES OF STAGE 2 LOSSES AND SENSITIVITY CONSTRAINTS

In the following, we assume that we obtained samples $\widetilde{u} = (\widetilde{u}_{x,a}^{(j)})_{j=1}^k \overset{\text{i.i.d.}}{\sim} \mathcal{N}(0_{d_y}, I_{d_y})$ and $\xi = (\xi_{x,a}^{(j)})_{j=1}^k \overset{\text{i.i.d.}}{\sim} \text{Bernoulli}(\mathbb{P}_{\text{obs}}(a \mid x))$.

### E.1.1   STAGE 2 LOSSES

Here, we provide our estimators $\hat{\mathcal{L}}_2(\eta, \widetilde{u}, \xi)$ of the Stage 2 loss $\mathcal{L}_2(\eta)$. We consider three different causal queries: (i) expectations, (ii) set probabilities, and (iii) quantiles.

**Expectations:** Expectations correspond to setting $\mathcal{F}(\mathbb{P}) = \mathbb{E}[X]$. Then, we can estimate our Stage 2 loss via the empirical mean, i.e.,

$$\hat{\mathcal{L}}_2(\eta, \widetilde{u}, \xi) = \frac{1}{k} \sum_{i=1}^n \sum_{j=1}^k f_{g_{\theta_{\text{opt}}^*}(x_i, a_i)}^* \left( (1 - \xi_{x_i, a_i}^{(j)}) \widetilde{f}_{\widetilde{g}_\eta(x_i, a_i)}(\widetilde{u}_{x_i, a_i}^{(j)}) + \xi_{x_i, a_i}^{(j)} \widetilde{u}_{x_i, a_i}^{(j)} \right). \tag{73}$$

**Set probabilities:** Here we consider queries of the form $\mathcal{F}(\mathbb{P}) = \mathbb{P}(X \in \mathcal{S})$ for some set $S \subseteq \mathcal{Y}$. We first define the log-likelihood

$$\ell\left(\eta, \widetilde{u}_{x_i, a_i}^{(j)}, \xi_{x_i, a_i}^{(j)}\right) = \mathbb{P}\left( f_{g_{\theta_{\text{opt}}^*}(x_i, a_i)}^* \left( (1 - \xi_{x_i, a_i}) \widetilde{f}_{\widetilde{g}_\eta(x_i, a_i)}(\widetilde{U}) + \xi_{x_i, a_i} \widetilde{U} \right) = \\ f_{g_{\theta_{\text{opt}}^*}(x_i, a_i)}^* \left( (1 - \xi_{x_i, a_i}^{(j)}) \widetilde{f}_{\widetilde{g}_\eta(x_i, a_i)}(\widetilde{u}_{x_i, a_i}^{(j)}) + \xi_{x_i, a_i}^{(j)} \widetilde{u}_{x_i, a_i}^{(j)} \right) \right), \tag{74}$$

which corresponds to the log-likelihood of the shifted distribution (under Stage 2) at the point that is obtained from plugging the Monte Carlo samples $\widetilde{u}_{x_i, a_i}^{(j)}$ and $\xi_{x_i, a_i}^{(j)}$ into the CNFs. We then optimize Stage 2 by maximizing this log-likelihood only at points in $\mathcal{S}$. That is, the corresponding Monte Carlo estimator of the Stage 2 loss is

$$\hat{\mathcal{L}}_2(\eta, \widetilde{u}, \xi) = \sum_{i=1}^n \sum_{j=1}^k \ell\left(\eta, \widetilde{u}_{x_i, a_i}^{(j)}, \xi_{x_i, a_i}^{(j)}\right) \\ \mathbb{1}\left\{ f_{g_{\theta_{\text{opt}}^*}(x_i, a_i)}^* \left( (1 - \xi_{x_i, a_i}^{(j)}) \widetilde{f}_{\widetilde{g}_\eta(x_i, a_i)}(\widetilde{u}_{x_i, a_i}^{(j)}) + \xi_{x_i, a_i}^{(j)} \widetilde{u}_{x_i, a_i}^{(j)} \right) \in \mathcal{S} \right\}. \tag{75}$$

Here, we only backpropagate through the log-likelihood to obtain informative (non-zero) gradients.

**Quantiles:** We consider quantiles of the form $\mathcal{F}(\mathbb{P}) = F_X^{-1}(q)$, where $F_X$ is the c.d.f. corresponding to $\mathbb{P}$ and $q \in (0, 1)$. For this, we can use the same Stage 2 loss as in Eq. (75) by defining the set $\mathcal{S} = \left\{ y \in \mathcal{Y} \mid y \leq \hat{F}_{ij}^{-1}(q) \right\}$ where $\hat{F}_{ij}^{-1}$ is empirical c.d.f. corresponding to

$$\left\{ f_{g_{\theta_{\text{opt}}^*}(x_i, a_i)}^* \left( (1 - \xi_{x_i, a_i}^{(j)}) \widetilde{f}_{\widetilde{g}_\eta(x_i, a_i)}(\widetilde{u}_{x_i, a_i}^{(j)}) + \xi_{x_i, a_i}^{(j)} \widetilde{u}_{x_i, a_i}^{(j)} \right) \right\}_{j=1}^k.$$

### E.1.2   SENSITIVITY CONSTRAINTS

Here, we provide our estimators $\hat{\mathcal{D}}_{x,a}(\eta, \widetilde{u})$ of the sensitivity constraint $\mathcal{D}_{x,a}(\mathbb{P}(U \mid x), \mathbb{P}(U \mid x, a))$. We consider the three sensitivity models from the main paper: (i) MSM, (ii) $f$-sensitivity models, and (iii) Rosenbaum's sensitivity model.

**MSM:** We define

$$\hat{\rho}(x, u, a, \eta) = \frac{1}{1 - \mathbb{P}(a \mid x)} \left( \frac{\mathbb{P}\left( (1 - \xi_{x,a}) \widetilde{f}_{\widetilde{g}_\eta(x, a)}(\widetilde{U}) + \xi_{x,a} \mathbb{P}(\widetilde{U} = u) \right)}{\mathbb{P}(\widetilde{U} = u)} - \mathbb{P}(a \mid x) \right). \tag{76}$$

Then, our estimator for the MSM constraint is

$$\hat{\mathcal{D}}_{x,a}(\eta, \widetilde{u}) = \max \left\{ \max_{u \in \widetilde{u}} \hat{\rho}(x, u, a, \eta), \ \max_{u \in \widetilde{u}} \hat{\rho}(x, u, a, \eta)^{-1} \right\}. \tag{77}$$

$f$-**sensitivity models:** Our estimator for the $f$-sensitivity constraint is

$$\hat{\mathcal{D}}_{x,a}(\eta, \widetilde{u}) = \max \left\{ \frac{1}{k} \sum_{j=1}^{k} f(\hat{\rho}(x, \widetilde{u}_{x,a}^{(j)}, a, \eta)), \ \frac{1}{k} \sum_{j=1}^{k} f(\hat{\rho}(x, \widetilde{u}_{x,a}^{(j)}, a, \eta)^{-1}) \right\}. \tag{78}$$

**Rosenbaum's sensitivity model:** We define

$$\hat{\rho}(x, u_1, u_2, a, \eta) = \frac{\mathbb{P}(\widetilde{U} = u_1)}{\mathbb{P}(\widetilde{U} = u_2)} \left( \frac{\mathbb{P}(\widetilde{U} = u_2)\mathbb{P}(a \mid x) - \mathbb{P}\left( (1 - \xi_{x,a})\widetilde{f}_{\widetilde{g}_\eta(x,a)}(\widetilde{U}) + \xi_{x,a}\mathbb{P}(\widetilde{U} = u_2) \right)}{\mathbb{P}(\widetilde{U} = u_1)\mathbb{P}(a \mid x) - \mathbb{P}\left( (1 - \xi_{x,a})\widetilde{f}_{\widetilde{g}_\eta(x,a)}(\widetilde{U}) + \xi_{x,a}\mathbb{P}(\widetilde{U} = u_1) \right)} \right). \tag{79}$$

Then, our estimator is

$$\hat{\mathcal{D}}_{x,a}(\eta, \widetilde{u}) = \max \left\{ \max_{u_1, u_2 \in \widetilde{u}} \hat{\rho}(x, u_1, u_2, a, \eta), \ \max_{u_1, u_2 \in \widetilde{u}} \hat{\rho}(x, u_1, u_2, a, \eta)^{-1} \right\}. \tag{80}$$

### E.2 FULL LEARNING ALGORITHM

Our full learning algorithm for Stage 1 and Stage 2 is shown in Algorithm 1. For Stage 2, we use our Monte-Carlo estimators described in the previous section in combination with the augmented lagrangian method to incorporate the sensitivity constraints. For details regarding the augmented lagrangian method, we refer to Nocedal & Wright (2006), chapter 17.

**Reusability:** Using CNFs instead of unconditional normalizing flows allows us to compute bounds $Q_{\mathcal{M}}^+(x, a)$ and $Q_{\mathcal{M}}^-(x, a)$ without the need to retrain our model for different $x \in \mathcal{X}$ and $a \in \mathcal{A}$. In particular, we can simultaneously compute bounds for averaged queries or differences *without* retraining (see Appendix D). Furthermore, Stage 1 is independent of the sensitivity model, which means that we can reuse our fitted Stage 1 CNF for different sensitivity models and sensitivity parameters $\Gamma$, and only need to retrain the Stage 2 CNF.

**Analytical potential outcome density:** Once our model is trained, we can not only compute the bounds via sampling but also the analytical form (by using the density transformation formula) of the potential outcome density that gives rise to that bound. Fig. 8, shows an example. This makes it possible to perform sensitivity analysis for the potential outcome density itself, i.e., analyzing the "distribution shift due to intervention".

---

**Algorithm 1:** Full learning algorithm for NEURALCSA

---

**Input** : causal query $Q(x, a)$, GTSM $\mathcal{M}$, and obs. dataset $\mathcal{D} \sim \mathbb{P}_{\text{obs}}$
         epoch numbers $n_0$, $n_1$, $n_2$; batch size $n_b$; learning rates $\gamma_0$, $\gamma_1$, and $\alpha > 1$

**Output:** learned parameters $\theta_{\text{opt}}$ and $\eta_{\text{opt}}$ of Stage 1 and Stage 2

`// Stage 1`

$\mathbb{P}^*(U \mid x, a) \leftarrow$ fixed probability distribution on $\mathcal{U} \subseteq \mathbb{R}^{d_u = d_y}$

Initialize $\theta^{(1)}$ and $U \sim \mathcal{N}(0_{d_y}, I_{d_y})$

**for** $i \in \{1, \ldots, n_0\}$ **do**

    $(x, a, y) \leftarrow$ batch of size $n_b$

    $\mathcal{L}_1(\theta) \leftarrow \sum_{(x,a,y)} \log \mathbb{P}(f^*_{g^*_\theta(x,a)}(U) = y)$

    $\theta^{(i+1)} \leftarrow$ optimization step of $\mathcal{L}_1(\theta^{(i)})$ w.r.t. $\theta^{(i)}$ with learning rate $\gamma_0$

**end**

$\theta_{\text{opt}} \leftarrow \theta^{(n_0)}$

`// Stage 2`

Initialize $\eta^{(1)}$, $\lambda^{(1)}$, and $\mu^{(1)}$

**for** $i \in \{1, \ldots, n_1\}$ **do**

    **for** $\ell \in \{1, \ldots, n_2\}$ **do**

        $\eta_1^{(i)} \leftarrow \eta^{(i)}$

        $(x, a) \leftarrow$ batch of size $n_b$

        $\widetilde{u} \leftarrow (\widetilde{u}^{(j)}_{x,a})^k_{j=1} \overset{i.i.d.}{\sim} \mathcal{N}(0_{d_y}, I_{d_y})$

        $\xi \leftarrow (\xi^{(j)}_{x,a})^k_{j=1} \overset{i.i.d.}{\sim} \text{Bernoulli}(\mathbb{P}_{\text{obs}}(a \mid x))$

        $s_{x,a}(\eta) \leftarrow \Gamma - \hat{\mathcal{D}}_{x,a}(\eta, \widetilde{u})$

        $\mathcal{L}(\eta, \lambda, \mu) \leftarrow \hat{\mathcal{L}}_2(\eta, \widetilde{u}, \xi) - \sum_{(x,a)} \lambda_{x,a} s_{x,a}(\eta) + \frac{\mu}{2} s^2_{x,a}(\eta)$

        $\eta^{(i)}_{\ell+1} \leftarrow$ optimization step of $\mathcal{L}(\eta^{(i)}_\ell, \lambda^{(i)}, \mu^{(i)})$ w.r.t. $\eta^{(i)}_\ell$ with learning rate $\gamma_1$

    **end**

    $\eta^{(i+1)} \leftarrow \eta^{(i)}_{n_2}$

    $\lambda^{(i+1)}_{x,a} \leftarrow \max\left\{0, \lambda^{(i)}_{x,a} - \mu^{(i)} s_{x,a}(\eta^{(i+1)})\right\}$

    $\mu^{(i+1)} \leftarrow \alpha \mu^{(i)}$

**end**

$\eta_{\text{opt}} \leftarrow \eta^{(n_1)}$

---

### E.3 FURTHER DISCUSSION OF OUR LEARNING ALGORITHM

**Non-neural alternatives:** Note that our two-stage procedure is agnostic to the estimators used, which, in principle, allows for non-neural instantiations. However, we believe that our neural instantiation (NEURALCSA) offers several advantages over possible non-neural alternatives:

1. Solving Stage 1: Conditional normalizing flows (CNFs) are a natural choice for Stage 1 because they are designed to learn an invertible function $f^*_{x,a} \colon \mathcal{U} \to \mathcal{Y}$ that satisfies $\mathbb{P}_{\text{obs}}(Y \mid x, a) = \mathbb{P}^*(f^*_{x,a}(U) \mid x, a)$. In particular, CNFs allow for inverting $f^*_{x,a}$ analytically, which enables tractable optimization of the log-likelihood (see Appendix F). In principle, $f^*_{x,a}$ could also be obtained by estimating the conditional c.d.f. $\hat{F}(y \mid x, a)$ using some arbitrary estimator and then leveraging the inverse transform sampling theorem, which states that we can choose $f^*_{x,a} = \hat{F}^{-1}(\cdot \mid, x, a)$ whenever we fix $\mathbb{P}^*(U \mid x, a)$ to be uniform. However, this approach only works for one-dimensional $Y$ and requires inverting the estimated $\hat{F}(y \mid x, a)$ numerically.

2. Solving Stage 2: Stage 2 requires to optimize the causal query $\mathcal{F}\left(\mathbb{P}(f^*_{x,a}(U) \mid x)\right)$ over a latent distribution $\mathbb{P}(U \mid x)$, where $f^*_{x,a}$ is learned in Stage 1. In NEURALCSA, we achieve this by fitting a second CNF in the latent space $\mathcal{U}$, which we then concatenate with the CNF from Stage 1 to backpropagate through both CNFs. Standard density estimators are not applicable in Stage 2 because we fit a density in the latent space to optimize a causal query that is dependent on Stage 1, and not a standard log-likelihood. While there may exist

      non-neural alternatives to solve the optimization in Stage 2, this goes beyond the scope of our paper, and we leave this for future work.

3. Analytical interventional density: Using NFs in both Stages 1 and 2 enables us to obtain an analytical interventional once NeuralCSA is fitted. Hence, we can perform sensitivity analysis for the whole interventional density (see Fig. 7 and 8), without the need for Monte Carlo approximations through sampling.

4. Universal density approximation: CNFs are universal density approximators, which means that we can account for complex (e.g., multi-modal, skewed) observational distributions.

**Complexity of NEURALCSA compared to closed-form solutions:** Stage 1 requires fitting a CNF to estimate the observational distribution $\mathbb{P}_{\mathrm{obs}}(Y \mid x, a)$. This step is also necessary for closed-form bounds (e.g., for the MSM), where such closed-form bounds depend on the observational distribution (Frauen et al., 2023b). This renders the complexity in terms of implementation choices equivalent to Stage 1.

In Stage 2, closed-form solutions under the MSM allow for computing bounds directly using the observational distribution, NEURALCSA fits an additional NF that is training using Algorithm 1. Hence, additional hyperparameters are the hyperparameters of the second NF as well as the learning rates of the augmented Lagrangian method (used to incorporate the sensitivity constraint). Hence, we recommend using NEURALCSA as a method for causal sensitivity analysis whenever bounds are not analytically tractable. In our experiments, we observed that the training of NeuralCSA was very stable, as indicated by a low variance over different runs (see, e.g., Fig. 5).

**Existence of a global optimum:** A sufficient condition for the existence of a global solution in Eq. (4) is the continuity of the objective/causal query as well as the compactness of the constraint set. Continuity holds for many common causal queries such as the expectation. The compactness of the constraint set depends on the properties functional $\mathcal{D}_{x,a}$, i.e., the choice of the sensitivity model. The existence of global solutions has been shown for many sensitivity models from the literature, e.g., MSM (Dorn et al., 2022) and $f$-sensitivity models (Jin et al., 2022). Note that, in Theorem 1, we do not assume the existence of a global solution. In principle, our two-stage procedure is valid even if a global solution to Eq. (5) does not exist. In this case, we can apply our Stage 2 learning algorithm (Algorithm 1) until convergence and obtain an approximation of the desired bound, even if it is not contained in the constraint set.

# F    DETAILS ON IMPLEMENTATION AND HYPERPARAMETER TUNING

**Stage 1 CNF**: We use a conditional normalizing flows (CNF) (Winkler et al., 2019) for stage 1 of NEURALCSA. Normalizing flows (NFs) model a distribution $\mathbb{P}(Y)$ of a target variable $Y$ by transforming a simple base distribution $\mathbb{P}(U)$ (e.g., standard normal) of a latent variable $U$ through an invertible transformation $Y = f_{\hat{\theta}}(U)$, where $\hat{\theta}$ denotes parameters (Rezende & Mohamed, 2015). In order to estimate the *conditional* distributions $\mathbb{P}(Y \mid x, a)$, CNFs define the parameters $\hat{\theta}$ as an output of a *hyper network* $\hat{\theta} = g_\theta(x, a)$ with learnable parameters $\theta$. The conditional log-likelihood can be written analytically as

$$\log\left(\mathbb{P}(f_{g_\theta(x,a)}(U) = y)\right) \overset{(*)}{=} \log\left(\mathbb{P}\left(U = f_{g_\theta(x,a)}^{-1}(y)\right)\right) + \log\left(\det\left(\frac{\mathrm{d}}{\mathrm{d}y} f_{g_\theta(x,a)}^{-1}(y)\right)\right), \quad (81)$$

where $(*)$ follows from the change-of-variables theorem for invertible transformations.

**Stage 2 CNF**: As in Stage 1, we use a CNF that transforms $U = \widetilde{f}_{\hat{\eta}}(\widetilde{U})$, where $\hat{\eta} = \widetilde{g}_\eta(x, a)$ with learnable parameters $\eta$. The conditional log-likelihood can be expressed analytically via

$$\log\left(\mathbb{P}(\widetilde{f}_{\widetilde{g}_\theta(x,a)}(\widetilde{U}) = u)\right) = \log\left(\mathbb{P}\left(\widetilde{U} = \widetilde{f}_{\widetilde{g}_\theta(x,a)}^{-1}(u)\right)\right) + \log\left(\det\left(\frac{\mathrm{d}}{\mathrm{d}u} \widetilde{f}_{\widetilde{g}_\theta(x,a)}^{-1}(u)\right)\right). \quad (82)$$

In our implementation, we use autoregressive neural spline flows. That is, we model the invertible transformation $f_\theta$ via a spline flow as described in Dolatabadi et al. (2020). We use an autoregressive neural network for the hypernetwork $g_\eta(\mathbf{x}, \mathbf{m}, \mathbf{a})$ with 2 hidden layers, ReLU activation functions, and linear output. For training, we use the Adam optimizer (Kingma & Ba, 2015).

**Propensity scores:** The estimation of the propensity scores $\mathbb{P}(a \mid \mathbf{x})$ is a standard binary classification problem. We use feed-forward neural networks with 3 hidden layers, ReLU activation functions, and softmax output. For training, we minimize the standard cross-entropy loss by using the Adam optimizer (Kingma & Ba, 2015).

**Hyperparameter tuning:** We perform hyperparameter tuning for our propensity score models and Stage 1 CNFs. The tunable parameters and search ranges are shown in Table 4. Then, we use the same optimally trained propensity score models and Stage 1 CNFs networks for all Stage 2 models and closed-form solutions (in Fig. 5). For the Stage 2 CNFs, we choose hyperparameters that lead to a stable convergence of Alg. 1, while ensuring that the sensitivity constraints are satisfied. For reproducibility purposes, we report the selected hyperparameters as *.yaml* files.[9]

Table 4: Hyperparameter tuning details.

| MODEL | TUNABLE PARAMETERS | SEARCH RANGE |
|---|---|---|
| Stage 1 CNF | Epochs | 50 |
| | Batch size | 32, 64, 128 |
| | Learning rate | 0.0005, 0.001, 0.005 |
| | Hidden layer size (hyper network) | 5, 10, 20, 30 |
| | Number of spline bins | 2, 4, 8 |
| Propensity network | Epochs | 30 |
| | Batch size | 32, 64, 128 |
| | Learning rate | 0.0005, 0.001, 0.005 |
| | Hidden layer size | 5, 10, 20, 30 |
| | Dropout probability | 0, 0.1 |

---

[9] Code is available at https://github.com/DennisFrauen/NeuralCSA.

## G   DETAILS REGARDING DATASETS AND EXPERIMENTS

We provide details regarding the datasets we use in our experimental evaluation in Sec. 6.

### G.1   SYNTHETIC DATA

**Binary treatment:** We simulate an observed confounder $X \sim \text{Uniform}[-1, 1]$ and define the observed propensity score as $\pi(x) = \mathbb{P}_{\text{obs}}(A = 1 \mid x) = 0.25 + 0.5\,\sigma(3x)$, where $\sigma(\cdot)$ denotes the sigmoid function. Then, we simulate an unobserved confounder

$$U \mid X = x \sim \text{Bernoulli}\left( p = \frac{(\Gamma - 1)\pi(x) + 1}{\Gamma + 1} \right) \tag{83}$$

and a binary treatment

$$A = 1 \mid X = x, U = u \sim \text{Bernoulli}\left( p = u\pi(x)s^+(x, a) + (1 - u)\pi(x)s^-(x, a) \right), \tag{84}$$

where $s^+(x, a) = \frac{1}{(1 - \Gamma^{-1})\pi(x) + \Gamma^{-1}}$ and $s^-(x, a) = \frac{1}{(1 - \Gamma)\pi(x) + \Gamma}$.

Finally, we simulate a continuous outcome

$$Y = (2A - 1)X + (2A - 1) - 2\sin(2(2A - 1)X) - 2(2U - 1)(1 + 0.5X) + \varepsilon, \tag{85}$$

where $\varepsilon \sim \mathcal{N}(0, 1)$.

The data-generating process is constructed so that $\frac{\mathbb{P}(A = 1 \mid x, u)}{\pi(x)} = us^+(x, a) + (1 - u)s^-(x, a)$ or, equivalently, that $\text{OR}(x, u) = u\Gamma + (1 - u)\Gamma^{-1}$. Hence, the full distribution follows an MSM with sensitivity parameter $\Gamma$. Furthermore, by Eq. (83), we have

$$\mathbb{P}(A = 1 \mid x, 1)\mathbb{P}(U = 1 \mid x) + \mathbb{P}(A = 1 \mid x, 0)\mathbb{P}(U = 0 \mid x) = \pi(x) = \mathbb{P}_{\text{obs}}(A = 1 \mid x) \tag{86}$$

so that $\mathbb{P}$ induces $\mathbb{P}_{\text{obs}}(A = 1 \mid x)$.

**Continuous treatment:** We simulate an observed confounder $X \sim \text{Uniform}[-1, 1]$ and independently a binary unobserved confounder $U \sim \text{Bernoulli}(p = 0.5)$. Then, we simulate a continuous treatment via

$$A \mid X = x, U = u \sim \text{Beta}(\alpha, \beta) \text{ with } \alpha = \beta = 2 + x + \gamma(u - 0.5), \tag{87}$$

where $\gamma$ is a parameter that controls the strength of unobserved confounding. In our experiments, we chose $\gamma = 2$. Finally, we simulate an outcome via

$$Y = A + X \exp(-XA) - 0.5(U - 0.5)X + (0.5X + 1) + \varepsilon, \tag{88}$$

where $\varepsilon \sim \mathcal{N}(0, 1)$. Note that the data-generating process does *not* follow a (continuous) MSM.

**Oracle sensitivity parameters $\Gamma^*$:** We can obtain Oracle sensitivity parameters $\Gamma^*(x, a)$ for each sample with $X = x$ and $A = a$ by simulating from our previously defined data-generating process to estimate the densities $\mathbb{P}(u \mid x, a)$ and $\mathbb{P}(u \mid x)$ and subsequently solve for $\Gamma^*(x, a)$ in the GTSM equations from Lemma 3. By definition, $\Gamma^*(x, a)$ is the smallest sensitivity parameter such that the corresponding sensitivity model is guaranteed to produce bounds that cover the ground-truth causal query. For binary treatments, it holds $\Gamma^*(x, a) = \Gamma^*$, i.e., the oracle sensitivity parameter does not depend on $x$ and $a$. For continuous treatments, we choose $\Gamma^* = \Gamma^*(a) = \int \Gamma^*(x, a)\mathbb{P}(x)\,\mathrm{d}x$ in Fig. 6.

### G.2   SEMI-SYNTHETIC DATA

We obtain covariates $X$ and treatments $A$ from MIMIC-III (Johnson et al., 2016) as described in the paragraph regarding real-world data below. Then we learn the observed propensity score $\hat{\pi}(x) = \hat{\mathbb{P}}(A = 1 \mid x)$ using a feed-forward neural network with three hidden layers and ReLU activation function. We simulate a uniform unobserved confounder $U \sim \mathcal{U}[0, 1]$. Then, we define a weight

$$w(X, U) = \mathbb{1}\left\{ \gamma \geq 2 - \frac{1}{\hat{\pi}(X)} \right\}(\gamma + 2U(1 - \gamma)) + \tag{89}$$

$$\mathbb{1}\left\{ \gamma < 2 - \frac{1}{\hat{\pi}(X)} \right\}\left( 2 - \frac{1}{\hat{\pi}(X)} + 2U\left( \frac{1}{\hat{\pi}(X)} - 1 \right) \right) \tag{90}$$

and simulate synthetic binary treatments via

$$A = 1 \mid X = x, U = u \sim \text{Bernoulli}\left(p = w(x,u)\hat{\pi}(x)\right). \tag{91}$$

Here, $\gamma$ is a parameter controlling the strength of unobserved confounding, which we set to $0.25$. The data-generating process is constructed in a way such that the full propensity $\mathbb{P}(A = 1 \mid X = x, U = u)$ induces the (estimated) observed propensity $\hat{\pi}$ from the real-world data. We then simulate synthetic outcomes via

$$Y = (2A - 1)\left(\frac{1}{d_x + 1}\left(\left(\sum_{i=1}^{d_x} X_i\right) + U\right)\right) + \varepsilon, \tag{92}$$

where $\varepsilon \sim \mathcal{N}(0, 0.1)$. In our experiments, we use 90% of the data for training and validation, and 10% of the data for evaluating test performance. From our test set, we filter out all samples that satisfy either $\mathbb{P}(A = 1 \mid x) < 0.3$ or $\mathbb{P}(A = 1 \mid x) > 0.7$. This is because these samples are associated with large empirical uncertainty (low sample sizes). In our experiments, we only demonstrate the validity of our NEURALCSA bounds in settings with low empirical uncertainty.

**Oracle sensitivity parameters $\Gamma^*$:** Similar to our fully synthetic experiments, we first obtain the Oracle sensitivity parameters $\Gamma^*(x, a)$ for each test sample with confounders $x$ and treatment $a$. We then take the median overall $\Gamma^*(x, a)$ of the test sample. By definition, NEURALCSA should then cover at least 50% of all test query values (see Table 2).

## G.3 REAL-WORLD DATA

We use the MIMIC-III dataset Johnson et al. (2016), which includes electronic health records from patients admitted to intensive care units. We use a preprocessing pipeline (Wang et al., 2020) to extract patient trajectories with 8 hourly recorded patient characteristics (heart rate, sodium blood pressure, glucose, hematocrit, respiratory rate, age, gender) and a binary treatment indicator (mechanical ventilation). We then sample random time points for each patient trajectory and define the covariates $X \in \mathbb{R}^8$ as the past patient characteristics averaged over the previous 10 hours. Our treatment $A \in \{0, 1\}$ is an indicator of whether mechanical ventilation was done in the subsequent 10-hour time. Finally, we consider the final heart rate and blood pressure averaged over 5 hours as outcomes. After removing patients with missing values and outliers (defined by covariate values smaller than the corresponding 0.1th percentile or larger than the corresponding 99.9th percentile), we obtain a dataset of size $n = 14719$ patients. We split the data into train (80%), val (10%), and test (10%).

# H  ADDITIONAL EXPERIMENTS

## H.1  ADDITIONAL TREATMENT COMBINATIONS FOR SYNTHETIC DATA

Here, we provide results for additional treatment values $a \in \{0.1, 0.9\}$ for the synthetic experiments with continuous treatment in Sec. 6. Fig. 9 shows the results for our experiment where we compare the NEURALCSA bounds under the MSM with (optimal) closed-form solutions. Fig. 10 shows the results of our experiment where we compare the bounds of different sensitivity models. The results are consistent with our observations from the main paper and show the validity of the bounds obtained by NEURALCSA,

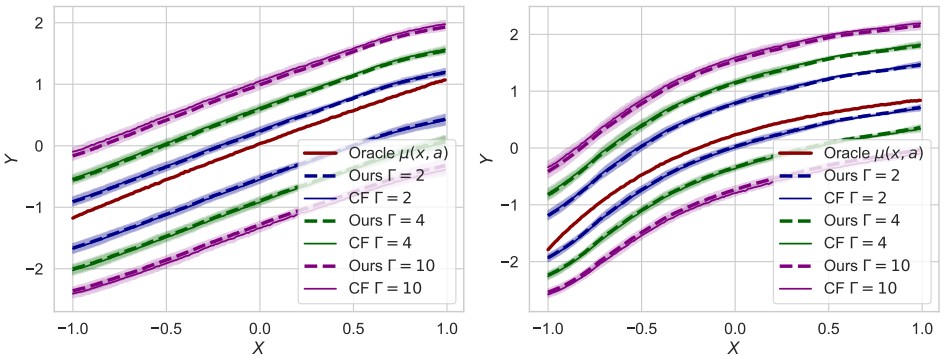

Figure 9: Validating the correctness of NEURALCSA (ours) by comparing with optimal closed-form solutions (CF) for the MSM in the synthetic continuous treatment setting. *Left: $a = 0.1$. Right: $a = 0.9$.* Reported: mean $\pm$ standard deviation over 5 runs.

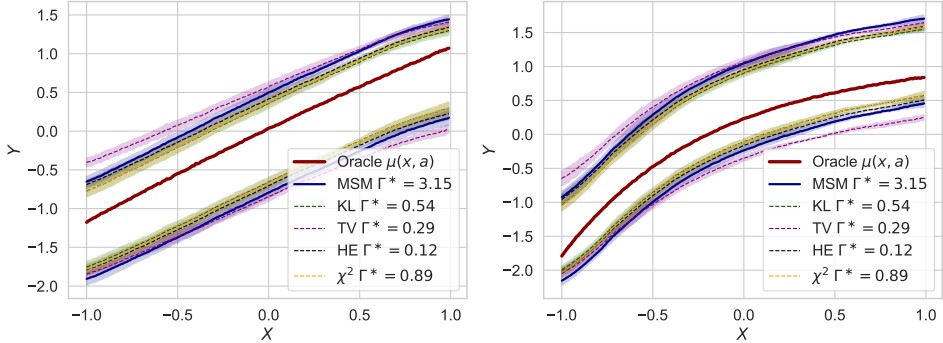

Figure 10: Confirming the validity of NEURALCSA bounds for various sensitivity models in the synthetic continuous treatment setting. *Left: $a = 0.1$. Right: $a = 0.9$.* Reported: mean $\pm$ standard deviation over 5 runs.

## H.2  DENSITIES FOR LOWER BOUNDS ON REAL-WORLD DATA

Here, we provide the distribution shifts for our real-world case study (Sec. 6), but for the lower bounds instead of the upper. The results are shown in Fig. 11. In contrast to the shifts for the upper bounds, increasing $\Gamma$ leads to a distribution shift *away* from the direction of the danger area, i.e., high heart rate and blood pressure.

## H.3  ADDITIONAL SEMI-SYNTHETIC EXPERIMENT

We provide additional experimental results using a semi-synthetic dataset based on the IHPD data Hill (2011). IHDP is a randomized dataset with information on premature infants. It was originally designed to estimate the effect of home visits from specialist doctors on cognitive test scores. For

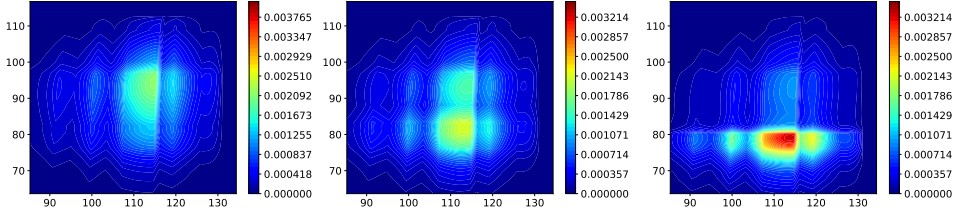

Figure 11: Contour plots of 2D densities obtained by NEURALCSA under an MSM. Here, we aim to learn a lower bound of the causal query $\mathbb{P}(Y_1(1) \geq 115, Y_2(1) \geq 90 \mid X = x_0)$ for a test patient $x_0$. *Left:* Stage 1/ observational distribution. *Center:* Stage 2, $\Gamma = 2$. *Right:* Stage 2, $\Gamma = 4$.

our experiment, we extract $d_x = 7$ covariates $X$ (birthweight, child's head circumference, number of weeks pre-term that the child was born, birth order, neo-natal health index, the mother's age, and the sex of the child) of $n = 985$ infants. Then, and define an observational propensity score as $\pi(x) = \mathbb{P}_{\text{obs}}(A = 1 \mid x) = 0.25 + 0.5\,\sigma(\frac{3}{d_x}\sum_{i=1}^{d_x} X_i)$. Then, similar as in Appendix G.2, we introduce unobserved confounding by generating synthetic treatments via

$$A = 1 \mid X = x, U = u \sim \text{Bernoulli}\left(p = w(x,u)\pi(x)\right), \tag{93}$$

where $w(x,u)$ is defined in Eq. (89) and $U \sim \mathcal{U}[0,1]$. We then generate synthetic outcomes via

$$Y = (2A - 1)\left(\frac{1}{d_x + 1}\left(\left(\sum_{i=1}^{d_x} X_i\right) + U\right)\right) + \varepsilon, \tag{94}$$

where $\varepsilon \sim \mathcal{N}(0,1)$.

In our experiments, we split the data into train (80%) and test set (20%). We verify the validity of our NEURALCSA bounds for CATE analogous to our experiments using the MIMIC (Sec. 6): For each sensitivity model (MSM, TV, HE, RB), we obtain the smallest oracle sensitivity parameter $\Gamma^*$ that guarantees coverage (i.e., satisfies the respective sensitivity constraint) for 50% of the test samples. Then, we plot the coverage and median interval length of the NEURALCSA bounds over the test set. The results are in Table 5. The results confirm the validity of NEURALCSA.

Table 5: Results for IHDP-based semi-synthetic data.

| Sensitivity model | Coverage | Interval length |
|---|---|---|
| MSM $\Gamma^* = 3.25$ | $0.92 \pm 0.03$ | $1.63 \pm 0.05$ |
| TV $\Gamma^* = 0.31$ | $0.71 \pm 0.28$ | $1.23 \pm 0.54$ |
| HE $\Gamma^* = 0.11$ | $0.70 \pm 0.14$ | $1.10 \pm 0.21$ |
| RB $\Gamma^* = 9.62$ | $0.56 \pm 0.15$ | $0.95 \pm 0.29$ |

Reported: mean $\pm$ standard deviation (5 runs).

## I    DISCUSSION ON LIMITATIONS AND FUTURE WORK

**Limitations:** NEURALCSA is a versatile framework that can approximate the bounds of causal effects in various settings. However, there are a few settings where (optimal) closed-form bounds exist (e.g., CATE for binary treatments under the MSM), which should be preferred when available. Instead, NEURALCSA offers a unified framework for causal sensitivity analysis under various sensitivity models, treatment types, and causal queries, and can be applied in settings where closed-form solutions have not been derived or do not exist (Table 1).

**Future work:** Our research hints at the broad applicability of NEURALCSA beyond the three sensitivity models that we discussed above (see also Appendix C). For future work, it would be interesting to conduct a comprehensive comparison of sensitivity models and provide practical recommendations for their usage. Future work may further consider incorporating techniques from semiparametric statistical theory to obtain estimation guarantees, robustness properties, and confidence intervals. Finally, we only provided identifiability results that hold in the limit of infinite data. It would be interesting to provide rigorous empirical uncertainty quantification for NEURALCSA, e.g., via a Bayesian approach. While in principle the bootstrapping approach from (Jesson et al., 2022) could be applied in our setting, this could be computationally infeasible for large datasets.

