# OpenReview forum: "A Neural Framework for Generalized Causal Sensitivity Analysis"
_ICLR.cc/2024/Conference — ICLR 2024 poster_

### Official Review · Reviewer_ijNf · 2023-10-21

**Soundness:** 3 good
**Presentation:** 4 excellent
**Contribution:** 3 good
**Rating:** 6
**Confidence:** 2

**Summary:**

This paper proposes a neural framework for generalized causal sensitivity analysis that aims to achieve lower and upper bounds for some causal queries when exact identification is not possible due to unobserved confounding. Specifically, the framework leverages two normalizing flows to encode the latent distributions that are compatible with the observed ones and optimize the causal quantities of interest and can be used to approximate previous sensitivity analysis models.

**Strengths:**

1. The writing is clear, and the paper is easy to follow.
2. The proposed neural network-based causal sensitivity analysis method is effective and general, which is compatible with previous studies like MSM and f-sensitivity models and can be easily extended to other models by modifying the constraints that specify the strength of unobserved confounding.

**Weaknesses:**

The experiments, to some extent, appear to be lacking in comprehensiveness. The semi-synthetic datasets utilized in this study are exclusively derived from the MIMIC-III dataset. The findings derived from this single source may not offer an adequate illustration of the model's performance.

**Questions:**

I noticed that the authors apply the augmented Lagrangian method to incorporate the sensitivity constraints in the optimization process. I wonder to what extent could the constraints be satisfied since the constraint now becomes a soft one that may well be violated. It is possible that the effective constraint parameter $\Gamma$ achieved by the optimization significantly deviates from the intended value, which may make the final bounds less useful. Additionally, I am concerned about the stability of training the normalizing flows, as instability in this aspect could further exacerbate the situation.

---

> ### Author Response · Authors · 2023-11-16
> **Response to Reviewer ijNf**
>
> Thank you for your helpful review! We took all your comments at heart and improved our paper as follows.
>
>
> ## Response to “Weaknesses”
>
>
>
> * Semi-synthetic experiments: We would like to emphasize that we not only evaluate NeuralCSA on semi-synthetic but also on various purely synthetic datasets (as common in causal inference [1, 2]). Nevertheless, we agree that additional experiments on semi-synthetic data are a great opportunity to demonstrate the effectiveness of our method.
>
>     **Action:** We provided **additional experimental** results using a semi-synthetic dataset based on the IHPD dataset [5] (see **new Appendix H.3**). The results confirm the validity of our bounds obtained via NeuralCSA.
>
>
>
> ## Response to “Questions”
>
>
>
> * Violation of sensitivity constraints: You are correct in that we leverage the Augmented Lagrangian method (AL) to incorporate the sensitivity constraints into Stage 2 of NeuralCSA (Algorithm 1). One reason why we chose AL is that **AL is specifically designed not to violate the constraint set** for a fixed constraint parameter. This is in contrast to other methods for constrained optimization (e.g., adding the sensitivity constraint as a penalty to the Stage 2 loss). Intuitively, the parameter $\mu^{(i)}$ is growing exponentially with respect to the number of epochs $i$, ensuring that the sensitivity violation $s_{x, a}$ converges to $0$. In practice, it is not necessary to let $\mu^{(i)}$ converge infinity to satisfy the constraint [3], which is what we also observe in our experiments. For a more comprehensive discussion on this and AL in general, we refer to [3]. Furthermore, the violation of the sensitivity constraint can be estimated during training, such that the number of epochs may be adjusted accordingly. In summary: **The achieved sensitivity parameter $\Gamma$ in our experiments always coincides with its intended value** (up to small numerical errors). As such, the bounds we obtain in our experiments directly correspond to the chosen sensitivity parameters.
>
>     **Action**: We added a clarification for the above point in our implementation paragraph (see Sec. 5.3).
>
> * Training of normalizing flows: It is true that our learning algorithm is more complex than comparable closed-form solutions (e.g., for MSM). However, even such closed-form solutions often require estimating a conditional distribution. As such, neural methods for conditional density estimation are common in the sensitivity analysis literature [1, 2]. We also would like to emphasize that we do not claim that NeuralCSA archives better performance than existing closed-form solutions. Rather, we propose our NeuralCSA as a method for causal sensitivity analysis whenever bounds are not analytically tractable (see Table 1). We argue that, in such settings, a certain complexity is expected by any approach that aims at deriving bounds. In our experiments, we observed that the training of NeuralCSA was very stable, as indicated by a low variance over different runs (see e.g., Fig. 4). Furthermore, there are a variety of methods available to improve training stability for conditional normalizing flows, e.g. noise regularization [4], which could be applied if necessary.
>
>     **Action:** We added a discussion to Appendix E.
>
>
>
> ## References
>
> [1] Frauen et al. (2023). “Sharp Bounds for Generalized Causal Sensitivity Analysis”. NeurIPS
>
> [2] Jesson et al. (2021). “Quantifying Ignorance in Individual-Level Causal-Effect Estimates under Hidden Confounding”. ICML.
>
> [3] Nocedal and Wright (2006). Numerical optimization. Springer series in operations research. Springer, New York, 2nd ed. edition, 2006. ISBN 0387303030
>
> [4] Rothfuss et al. (2020). “Noise Regularization for Conditional Density Estimation” 	arXiv:1907.08982
>
> [5] Hill (2011). “Bayesian nonparametric modeling for causal inference”. Journal of Computational and Graphical Statistics

---

### Official Review · Reviewer_MpUG · 2023-10-30

**Soundness:** 3 good
**Presentation:** 3 good
**Contribution:** 2 fair
**Rating:** 6
**Confidence:** 3

**Summary:**

This paper proposes a generalized sensitivity analysis framework that is compatible with many different sensitivity models, including the marginal sensitivity model, f-sensitivity model, and Rosenbaum's sensitivity model. The framework is suitable for different treatment types and different causal queries. The authors propose to learn the latent distribution shift with two separately trained conditional normalizing flows.

**Strengths:**

- The concise summary of sensitivity models enhances the paper's readability and flow.
- The authors introduce a novel learning strategy to model the latent distribution.
- Experiments with both synthetic and real-world data are used to demonstrate the validity and effectiveness of the proposed method.

**Weaknesses:**

- It is not very obvious how does the bounds the proposed framework compares with some existing works such as GMSM.
- The section 5.1 might be a little hard to follow. Please find some questions I have below.

**Questions:**

- Regarding the color coding in equation (4), does it indicate the parameters for the optimization problem? Does the right supremum also maximize over $P(U|x,a)$?
- I'm not fully understand the two-stage procedure. Could you to provide a more detailed explanation about replace the right supremum with
 fixed of $\mathbb{P}^*(U|x,a)$ and $\mathbb{P}^*(Y|x,u,a)$?
- In relation to the optimization problem presented in equation (5), are there specific constraints placed on the functional $\mathcal{D}_{x,a}$ to ensure that the global optimal can be achieved?

---

> ### Author Response · Authors · 2023-11-16
> **Response to Reviewer MpUG**
>
> Thank you for your positive review and your helpful comments!
>
>
> ## Response to “Weaknesses”
>
>
> * The GMSM (proposed by [1]) extends the standard MSM/ CMSM in two major ways: (i) It allows for _mediators_, thus enabling sensitivity analysis for direct/indirect/path-specific effects, and (ii) it allows for the incorporation of _weights_ into the sensitivity model that represent prior knowledge about the confounding structure. Point (i) is not relevant in our setting as we do not consider mediators. Extending NeuralCSA to settings with mediators may be an interesting direction for future work. For point (ii), we kindly refer you to Appendix C: Therein, **we show that the weighted GMSM can be reformulated as a GTSM, which makes NeuralCSA applicable**. For example, NeuralCSA could be used to perform sensitivity analysis for multiple outcomes under a weighted GMSM, which was previously not possible using the bounds in [1]. We would like to emphasize that we do not claim that NeuralCSA archives better performance than existing closed-form solutions (e.g., bounds for the GMSM in single-outcome settings). Rather, we propose our NeuralCSA as a method for causal sensitivity analysis whenever bounds are not analytically tractable (see Table 1). Furthermore, note that the bounds in [1] are not applicable to sensitivity models beyond MSM-type (e.g., $f$-sensitivity models, Rosenbaum’s sensitivity model), yet for which our NeuralCSA is applicable.
>
>
> ## Response to “Questions”
>
> * In Eq. (4), the coloring highlights the components of the joint probability distribution that appear in the causal query, i.e., the optimization objective. The optimization happens over the components that appear just below the suprema, that is, both $\mathbb{P}(U \mid x, a)$ and  ${\mathbb{P}(Y \mid x, u, a)\}$ for the right supremum, and $\{\mathbb{P}(U \mid x, a^\prime)\}_{a^\prime \neq a}$ for the left supremum.
>
>     **Action:** We added a clarification below Eq. (4) where we explain the choice behind the coloring.
>
> * Two-stage procedure: Theorem 1 has two major implications: (1) It is sufficient to fix the distributions $\mathbb{P}^\ast(U \mid x, a)$ and $\mathbb{P}^\ast(Y \mid x, u, a)$; and (2) it is sufficient to choose $\mathbb{P}^\ast(Y \mid x, u, a) = \delta(Y - f^\ast_{x, a}(u))$ as a delta-distribution induced by an invertible function $f^\ast_{x, a} \colon \mathcal{U} \to \mathcal{Y}$, which satisfies the data-compatibility constraint $\mathbb{P}(Y \mid x, a) =  \mathbb{P}^\ast(f^\ast_{x, a}(U) \mid x, a)$. Point (1) implies that we can fix the components in the right supremum of Eq.~(4) and only optimize over the left supremum. Point (2) implies that the causal query $\mathcal{F} \left(\int \mathbb{P}(Y \mid x, u, a) \mathbb{P}(u \mid x) du\right)$ reduces to $\mathcal{F} \left(\mathbb{P}(f^\ast_{x, a}(U \mid x) \right)$. Eq.(5) shows the corresponding Stage 2 optimization for discrete treatments (maximizing over $\mathbb{P}(u \mid x, A \neq a)$ to ensure data compatibility). For continuous treatments, we can directly maximize over $\mathbb{P}(u \mid x)$.
>
>     **Action:** We added a clarification to Sec. 5.2. Please let us know if you have further questions/suggestions regarding the clarification of our two-stage procedure.
>
> * Global optimum: A sufficient condition for the existence of a global solution in Eq.(5) is the continuity of the objective/causal query as well as the compactness of the constraint set. Continuity holds for many common causal queries such as the expectation. The compactness of the constraint set depends on the properties functional $\mathcal{D}_{x, a}$, i.e., the choice of the sensitivity model. Compactness (and thus the existence of global solutions) has been shown for many sensitivity models from the literature, e.g., MSM [2] and $f$-sensitivity models [3]. We would like to emphasize that, in Theorem 1, we do not assume the existence of a global solution. In principle, our two-stage procedure is valid even if a global solution to Eq.(5) does not exist. In this case, we can apply our Stage 2 learning algorithm (Algorithm 1) until convergence and obtain an approximation of the desired bound, even if it is not contained in the constraint set.
>
>     **Action:** We added the above discussion to Appendix E of our paper.
>
>
>
> ## References
>
> [1] Frauen et al. (2023). “Sharp Bounds for Generalized Causal Sensitivity Analysis”. NeurIPS
>
> [2] Dorn et al. (2022). “Doubly-Valid/Doubly-Sharp Sensitivity Analysis for Causal Inference with Unmeasured Confounding”. arXiv:2112.11449.
>
> [3] Jin et al. (2022). “Sensitivity analysis under the f-sensitivity models: Distributional robustness perspective”. arXiv:2203.04373.

---

### Official Review · Reviewer_9LA1 · 2023-11-01

**Soundness:** 3 good
**Presentation:** 3 good
**Contribution:** 3 good
**Rating:** 8
**Confidence:** 3

**Summary:**

The paper proposes NeuralCSA, a framework for performing causal sensitivity analysis, i.e., partially identification of a causal functional under assumptions on the unmeasured confounders. They propose a 2-stage training procedure, modeling the latent distributions using normalizing flows.

**Strengths:**

- The paper overall is well-written and easy to understand. I found the motivation and setup to be clear. I also appreciated comparisons to existing sensitivity models.
- The GTSM framework subsumes many of the existing sensitivity models and thus is more generally applicable. In principle, the framework also applies to arbitrary functionals (e.g., quantiles) of the interventional outcome distributions.
- The clarity of the two-stage procedure can be improved (see Weaknesses section), but overall, the procedure is simple and easy to follow. It is also nicely motivated using Theorem 1.

**Weaknesses:**

- I found Sec 5.1 and 5.2 difficult to read and I think clarity can be improved. What confused me initially was that you suggest fixing $P^*(U|x, a)$ but then the $\sup$ in Eq. 5 is also over the distributions $p(u|x, A)$. Reading it further, the sup is only for $A \neq a$ but I think clarifying that you only fix for the treatment $a$ that enters into $Q$ would be useful. Maybe this is obvious, but it will still make it easier to understand what is being optimized over in the $\sup$.
- It would also be nice to have some intuition of the proof of Theorem 1. Also, the invertible function $f^*$ would depend on the fixed $P^*$. Does certain distributions $P^*$ make it easier to determine $f^*$. In practice, how should you determine which $P^*$ to fix?

**Questions:**

See weaknesses section.

---

> ### Author Response · Authors · 2023-11-16
> **Response to Reviewer 9LA1 #1**
>
> Thank you for your positive evaluation of our paper! We took all your comments at heart and improved our paper accordingly.
>
> ## Response to “Weaknesses”
>
>
> * Thank you for pointing out the ambiguous notation in Eq. (5). You are correct in that we optimize the objective in Eq. (5) over the distribution $\mathbb{P}(u \mid x, A \neq a)$ for fixed $a$, and not directly over $\mathbb{P}(u \mid x)$. The variable $a$ is indeed the fixed treatment intervention that enters the causal query $Q$. Framing the optimization problem in this way ensures that the corresponding joint distribution $\mathbb{P}(u, a \mid x)$ induces the observational distribution $\mathbb{P}_\mathrm{obs}(a \mid x)$, which guarantees sharpness of our bounds without the need to incorporate additional constraints into our learning algorithm. If we would also optimize over distributions $\mathbb{P}(u \mid x)$ that are not compatible with the observational distribution, our bounds may become overly conservative.
>
>     **Action:** We added a clarification to Eq. (5) in order to improve the clarity of Sec. 5.1.

---

> ### Author Response · Authors · 2023-11-16
> **Response to Reviewer 9LA1 #2**
>
> * **Intuition for Theorem 1:** We agree that Sec. 5.1 and 5.2 can be improved by providing further intuition for Theorem 1. We believe that the following two aspects of Theorem 1 may need further clarification (for a rigorous proof, we refer to Appendix B.3):
>     * _Why is it sufficient to only consider invertible functions $f: \mathcal{U} \to \mathcal{Y}$, i.e., model $\mathbb{P}^\ast(Y \mid x, a, u) = \delta(Y - f(u))$  as a Dirac delta distribution?_ Here, it is helpful to have a closer look at Fig. 1: Intervening on the treatment $A$ causes a shift in the latent distribution of $U$, which then leads to a shifted interventional distribution $\mathbb{P}(Y(a) = y \mid x)$. The question is now: How can we obtain an interventional distribution that results in a maximal causal query (for the upper bound)? Let us consider a structural equation of the form $Y = g(U, \epsilon)$, where $\epsilon$ is some independent noise. Hence, the “randomness” (entropy) in $Y$ comes from both $U$ and $\epsilon$; however, the distribution shift only arises through $U$. Intuitively, the interventional distribution should be maximally shifted if $Y$ only depends on the unobserved confounder and not on independent noise, i.e., $g(U, \epsilon) = f(U)$ for some invertible function $f$. One may also think about this as achieving the maximal “dependence” (mutual information) between the random variables $U$ and $Y$. Note that any GTSM only restricts the dependence between $U$ and $A$, but not between $U$ and $Y$.
>     * _Why can we fix $\mathbb{P}^\ast(U \mid x, a)$ and $f$ without losing the ability to achieve the optimum in Eq. (4)?_ The basic idea is as follows: Let $\widetilde{\mathbb{P}}(\widetilde{U} \mid x, a)$ and $\widetilde{f}$ be optimal solutions to Eq. (4) for a potentially different latent variable $\widetilde{U}$. Then, we can define a mapping $t = {f}^{-1} \circ \widetilde{f} \colon \widetilde{U} \to U$ between latent spaces that transforms $\widetilde{\mathbb{P}}(\widetilde{U} \mid x, a)$ into our fixed $\mathbb{P}^\ast(U \mid x, a)$ (because both $\widetilde{f}$ and $f$ respect the observational distribution). Furthermore, we can use $t$ to push the optimal shifted distribution $\widetilde{\mathbb{P}}(\widetilde{U} \mid x)$ (under treatment intervention) to the latent variable $U$ (see Eq. (46)). In our proof in Appendix B.3. we show that this is sufficient to obtain a distribution  $\mathbb{P}^\ast$ that induces $\mathbb{P}^\ast(U \mid x, a)$ and $\mathbb{P}^\ast(Y \mid x, u, a) = \delta(Y - f(u))$ and which satisfies the sensitivity constraints. For the latter property, we require the sensitivity model to be “invariant” with respect to the transformation $t$. We formalize this via our transformation-invariance assumption (Definition 3).
>
>     **Choice of the fixed distribution $\mathbb{P}^\ast(U \mid x, a)$ in Stage 1:** You are correct in that the function $f^\ast_{x, a}$ depends on the choice of $\mathbb{P}^\ast(U \mid x, a)$ (we sightly abused notation here for improved readability). In practice, Stage 1 reduces to fitting a (conditional) normalizing flow (NF) to learn the observational distribution $\mathbb{P}_\mathrm{obs}(Y \mid x, a)$. **Hence, the problem of choosing  $\mathbb{P}^\ast(U \mid x, a)$ in Stage 1 of NeuralCSA is equivalent to choosing the latent distribution when fitting an NF**. Common choices from the literature are the standard Gaussian or uniform distribution [1]. In our implementation, we chose a standard Gaussian for the following reason: Stage 2 requires fitting a second NF to model a distribution in the latent space with the same support as $\mathbb{P}^\ast(U \mid x, a)$. By using a Gaussian, we avoid the need to hard-code support constraints into our Stage 2 NF.
>
>
>     **Action:** We extended our intuition paragraph in Sec. 5.2. and included some of the arguments provided above. Furthermore, we added an extensive discussion to Sec.B.3 in the appendix (see our proof of Theorem 1).
>
>
>
> ## References
>
> [1] Papamakarios et al. (2021). “Normalizing Flows for Probabilistic Modeling and Inference”. JMLR

---

> > ### Comment · Reviewer_9LA1 · 2023-11-22
> > **Response to authors**
> >
> > Thank you for your detailed responses. After reading this and the other reviews, I continue to be positive about the paper.

---

### Official Review · Reviewer_i97n · 2023-11-05

**Soundness:** 3 good
**Presentation:** 3 good
**Contribution:** 3 good
**Rating:** 6
**Confidence:** 3

**Summary:**

The authors provide a framework for generalized causal sensitivity analysis, an approach that subsumes three previous methods and provides additional advantages.

**Strengths:**

The key advantage of the authors' approach (according to the authors) is that it is much more widely applicable than any other single approach (which typically focus on a specific sensitivity model, treatment type, and causal query).

The paper is well-written, and it provides substantial background about existing methods for causal sensitivity analysis.

The experiments appear consistent with the existing literature, well-motivated, and useful.

**Weaknesses:**

The generality of the approach appears to come at a substantial cost in terms of complexity (with a corresponding potential for unexpected sources of error, bias, or misspecification). The single advantage over MSM appears to be allowing causal queries with multiple outcomes.

It is unclear the extent to which alternative (non-neural) implementations of the GTSM are possible. The paper would be improved by clearly describing what advantages the neural implementation provides over alternatives.

It seems somewhat odd to cite D'Amour 2019 (an excellent paper about a very specific topic) for the idea that "unobserved confounding often renders causal inference challenging." That has been known for more 50 years, going back at least to Reichenbach's common cause principle.

**Questions:**

Are alternative (non-neural) implementations of the GTSM possible? What advantages does the neural implementation provides over alternatives? How does the complexity (e.g., number of hyper-parameters and other implementation choices) of NeuralCSA compare to MSM?

---

> ### Author Response · Authors · 2023-11-16
> **Response to Reviewer i97n #1**
>
> Thank you for your positive review and your helpful comments. We improved our paper in the following ways:
>
> ## Response to “Weaknesses”
>
> * We agree that fitting NeuralCSA is more complex than using closed-form solutions. For practical usage, we recommend using closed-form solutions whenever available (e.g., for the MSM with a single outcome). However, for many settings, such closed-form solutions do **not** exist (see Table 1). This is where NeuralCSA (our framework) is of great value: it allows – for the first time – to obtain bounds that would otherwise be intractable.
>    We would also like to emphasize that our paper not only provides practical solutions for causal sensitivity analysis but also offers new perspectives for the development of future methods on partial identification/sensitivity analysis. To the best of our knowledge, we are the first to obtain bounds by explicitly learning the distribution shift in the latent unobserved confounders due to treatment intervention (Fig. 1). We suspect that this basic idea could also be applied in other settings where closed-form solutions are not available (e.g., mediation or IV settings), thus enabling potential follow-up works. **Action**: We expanded our discussion (Sec. 7) and included advice for when to employ NeuralCSA in practice.
>
> * Non-neural alternatives: This is an interesting question. First, our two-stage procedure is agnostic to the estimators used, which, in principle, allows for non-neural instantiations. However, we believe that there are important reasons why our **neural instantiation** (NeuralCSA, Sec. 4) has **several advantages** over possible non-neural alternatives:
>     * **Solving Stage 1**: Stage 1 requires learning an invertible function $f: \mathcal{U} \to \mathcal{Y}$ that transforms the fixed latent distribution $\mathbb{P}^\ast(U \mid x, a)$ (e.g., Gaussian) to the observed distribution $\mathbb{P}\mathrm{obs}(Y \mid x, a)$. Normalizing flows (NFs) are specifically designed for this task and, hence, are a natural choice for learning $f$. In particular, NFs allow for inverting $f$ analytically, which enables tractable optimization of the log-likelihood (see Appendix F). In principle, $f$ could also be obtained by estimating the conditional c.d.f. $\hat{F}(y \mid x, a)$ using some arbitrary estimator (e.g., non-neural [1]) and then leveraging the inverse transform sampling theorem, which states that we can choose $f = \hat{F}^{-1}$ whenever we fix $\mathbb{P}^\ast(U \mid x, a)$ to be uniform. However, this approach only works for one-dimensional $Y$ and requires inverting the estimated $\hat{F}(y \mid x, a)$ numerically.
>     * **Solving Stage 2**: Stage 2 requires to optimize the causal query $\mathcal{F} \left(\mathbb{P}(f^\ast_{x, a}(U) \mid x) \right)$ over a latent distribution $\mathbb{P}(U \mid x)$, where $f^\ast_{x, a}$ is learned in Stage 1. In NeuralCSA, we achieve this by fitting a second NF in the latent space $\mathcal{U}$, which we then concatenate with the NF from Stage 1 to backpropagate through both NFs. **Standard density estimators are not applicable** in Stage 2 because we fit a density **in the latent space** to optimize a causal query that is dependent on Stage 1 (and **not** a standard log-likelihood). While there may exist non-neural alternatives for solving the optimization in Stage 2, this goes beyond the scope of our paper, and we leave this for future work.
>     * **Analytical interventional density**: Using NFs in both Stages 1 and 2 enables us to obtain an analytical interventional once NeuralCSA is fitted. Hence, we can perform sensitivity analysis for the whole interventional density (see Fig. 6 and 7) in a scalable manner, without the need for Monte Carlo approximations through sampling.
>     * **Universal density approximation**: NFs are universal density approximators, which means that we can account for complex (e.g., multi-modal, skewed) observational distributions.
>
>     **Action:** We added a section Appendix E in which we discuss the advantages of our neural implementation and outline the possibility of non-neural alternatives.
>
> * D'Amour (2019) citation: We followed your suggestion and removed the citation. Instead, we now cite the causality book by Judea Pearl.

---

> ### Author Response · Authors · 2023-11-16
> **Response to Reviewer i97n #2**
>
> ## Response to “Questions”
>
> * Non-neural alternatives -> see our “Response to weaknesses”
> * Complexity of NeuralCSA vs the MSM (closed-form solutions):
>     * **Stage 1**: Stage 1 requires fitting an NF to estimate the observational distribution $\mathbb{P}\mathrm{obs}(Y \mid x, a)$. This step is also necessary for MSMs, where the closed-form solutions for bounds depend on the observational distribution (see [2]). Hence, the complexity of MSM (closed-form solutions) and Stage 1 of our NeuralCSCA is **equivalent**.
>     * **Stage 2**: While closed-form solutions under the MSM allow for computing bounds directly using the observational distribution, NeuralCSA fits an additional NF that is trained using Algorithm 1. Hence, Stage 2 introduces additional hyperparameters, namely: the hyperparameters of the second NF and the learning rates of the augmented Lagrangian method (used to incorporate the sensitivity constraint). In our experiments, NeuralCSA was able to effectively recover the closed-form bounds under the MSM despite the additional complexity due to Stage 2 (see Fig. 4). We would like to emphasize that we do not claim that NeuralCSA archives better performance than existing closed-form solutions. Rather, we propose our NeuralCSA as a method for causal sensitivity analysis whenever bounds are not analytically tractable.
>
>     **Action:** We added the above comparison to Appendix E of our paper. We also emphasized in our introduction and discussion that we primarily propose NeuralCSA for settings in which bounds are not analytically tractable.
>
> ## References
>
> [1] Chernozhukov et al. (2013). “Inference on counterfactual distributions”. Econometrica.
>
> [2] Frauen et al. (2023). “Sharp Bounds for Generalized Causal Sensitivity Analysis”. NeurIPS

---

### Author Response · Authors · 2023-11-16
**Response to all reviewers**

Thank you very much for the constructive evaluation of our paper and your helpful comments! We addressed all of them in the comments below. We uploaded a revised version of our paper, where we highlight key changes colored in red.

Our main improvements are the following:



* We provided additional intuition for Theorem 1 and our two-stage procedure, which should enhance the readability of Section 5.1. and 5.2.
* We clarified several points, including the notation in Eq.(4) and (5) as well as the complexity of NeuralCSA compared to closed-form solutions (see new Appendix E.3).
* We performed experiments on an additional semi-synthetic dataset based on the IHDP data.

We will incorporate all changes into the camera-ready version of our paper. Given these improvements, we are confident that our paper will be a valuable contribution to the causal machine learning literature and a good fit for ICLR 2024.

---

### Meta-Review · Area_Chair_eVwz · 2023-12-12

**Metareview:**

The authors develop a generalized causal sensitivity analysis that encompasses a large class of existing sensitivity models. This generality is achieved using normalizing flows to estimate a latent distribution in the sensitivity framework. Sensitivity analysis is a core part of checking observational causal effect estimates, so making it simpler to do via a unification has value. All of the reviewers were positive, though had lower confidence than typical, likely because of the specialized focus of the work and the technical background (around 4 pages) needed to carry it out for an ICLR audience.

**Justification For Why Not Higher Score:**

While the work is technically interesting, the experiments are limited and the writing moves slowly noting the heavy background needed for the ICLR community.

**Justification For Why Not Lower Score:**

Sensitivity analysis is important for causal inference. Placing them all under a general framework is fresh and has value for those working on observational causal inference.

---

### Decision · Program_Chairs · 2024-01-16

Accept (poster)